# Downscaling of climate change scenarios for a high resolution, site–specific assessment of drought stress risk for two viticultural regions with heterogeneous landscapes

Marco Hofmann[1], Claudia Volosciuk[2,3], Martin Dubrovský[4,5], Douglas Maraun[6] and Hans R. Schultz[1]

[1]Department of General and Organic Viticulture, Hochschule Geisenheim University, Geisenheim, 65366, Germany
[2]Science and Innovation Department, World Meteorological Organization, Geneva, CH-1211, Switzerland
[3]Deutscher Wetterdienst, Offenbach am Main, 63067, Germany
[4]Institute of Atmospheric Physics, The Czech Academy of Sciences, Prague, 141 00, Czech Republic
[5]Global Change Research Institute, The Czech Academy of Sciences, Brno, 603 00, Czech Republic
[6]Wegener Center for Climate and Global Change, University of Graz, Graz, 8010, Austria

*Correspondence to*: Marco Hofmann (marco.hofmann@hs-gm.de)

**Abstract.** Extended periods without precipitation, observed for example in Central Europe including Germany during the seasons from 2018 to 2020, can lead to water deficit and yield and quality losses for grape and wine production. Irrigation infrastructure in these regions to possibly overcome negative effects is largely non–existent. Regional climate models project changes of precipitation amounts and patterns, indicating an increase in frequency of occurrence of comparable situations in the future. In order to assess possible impacts of climate change on the water budget of grapevines, a water balance model was developed, which accounts for the large heterogeneity of vineyards with respect to their soil water storage capacity, evapotranspiration as a function of slope and aspect, and viticultural management practices. The model was fed with data from soil maps (soil type and plant available water capacity), a digital elevation model, the European Union (EU) vineyard–register, observed weather data and future weather data simulated by regional climate models and downscaled by a stochastic weather generator. This allowed conducting a risk assessment of the drought stress occurrence for the wine–producing regions Rheingau and Hessische Bergstraße in Germany on the scale of individual vineyard plots. The simulations showed that the risk for drought stress varies substantially between vineyard sites but might increase for steep–slope regions in the future. Possible adaptation measures depend highly on local conditions and are needed to make targeted use of the resource water, while an intense interplay of different wine-industry stakeholders, research, knowledge transfer, and local authorities will be required.

## 1 Introduction

Global mean temperature has increased and each decade since the 1980s has been warmer than any preceding one since 1850 (WMO, 2020). Accordingly, warming during the growing season (Apr–Oct, Northern Hemisphere, Oct–Apr, Southern Hemisphere) has been observed in all studied wine regions on several continents over the past 50–60 years (Schultz, 2000; Jones et al., 2005a; Webb et al., 2007, 2011; Santos et al., 2012). Changes in temperature have a pronounced effect on the geographical distribution of where grapevines can be grown (Kenny and Harrison, 1992; Jones et al., 2005b; Schultz and Jones, 2010; Santos et al., 2012), since this crop is highly responsive to environmental conditions (Sadras et al., 2012a). Within the existing production areas, where temperature conditions are generally in favour for cultivation, water shortage is probably the most dominant environmental constraint (Williams and Matthews, 1990). Even in moderate temperate climates, grapevines often face some degree of drought stress during the growing season (Morlat et al., 1992; Van Leeuwen and Seguin, 1994; Gaudillère et al., 2002; Gruber and Schultz, 2005; Gruber, 2012). Soil moisture has decreased across Europe since the beginning of the 20[th] century (Hanel et al., 2018) and in the most recent decade, the severity of drought events increased in southwestern Germany (Erfurt et al., 2020). This was in part a consequence of observed recent increases in potential evapotranspiration (Bormann, 2011; Hartmann et al., 2013; Schultz, 2017) and the natural variability of precipitation.

Despite some newly emerging wine regions at extreme latitudes to the north (Jones and Schultz, 2016), Germany's winegrowing regions are still at the northern fringe of economically important grape cultivation in Europe. Historically, viticulture is practiced only in climatically favorable regions, mostly located along river valleys on slopes or lowlands in the southwest of Germany. In many of these areas viticulture is the main socio–economic factor, determining the cultural landscape with steep slope regions additionally forming biodiversity hotspots (Jäger and Porten, 2018; Petermann et al., 2012). Mean annual precipitation is generally low in these steep slope regions (500–770 mm; 1971–2000; Ahr, Mittelrhein, Mosel, Nahe, Rheingau; DWD (Deutscher Wetterdienst), 2020) and available water capacity (*AWC*) of soils is very heterogeneous. Additionally, the percentage of vineyards with low *AWC* is relatively high (example Rheingau region; *AWC* < 125 mm for nearly 50 % of steep slope areas; Löhnertz et al., 2004) and the evaporative demand varies substantially within a growing region, because of different slopes and aspects of the vineyard plots (Hofmann and Schultz, 2015). Therefore, risk assessment of climate driven changes in soil and plant water budget needs to be on a fine scale in order to identify possible adaptation measures within growing regions. These measures may span changes in the selection of grapevine varieties and rootstocks, soil, cover crop or canopy management as well as the implementation of irrigation systems.

High spatial resolution predictions are a challenge in climate impact studies and mainly limited by the size of one grid box of regional climate models (RCMs). Although climatic conditions within a grid box may change from being suitable for vineyards to areas unsuitable for the cultivation of grapevines, climate change impact studies for European viticulture were often forced to be performed based on the spatial resolution of the underlying gridded climate model data. Santos et al. (2012) analyzed observed shifts of bioclimatic indices (mainly temperature related) by means of the E–OBS gridded data set and the connection with large scale atmospheric forcing. Projections of bioclimatic indices based on RCMs were analyzed by Malheiro et al. (2010) and Fraga et al. (2013), with the latter study also including possible changes in interannual variability. In terms of water supply, both studies projected a strong decrease of water availability for the Mediterranean basin but their projections differed for Central Europe ranging from a slight decrease (Fraga et al., 2013) to an increase (Malheiro et al., 2010). More specific regional aspects were analyzed by Santos et al. (2013) for the future of wine production in the Douro Valley (Portugal), and Moriondo et al. (2010) for expected changes in the premium wine quality area of Tuscany at a fine spatial resolution (1 km x 1 km, based on downscaling climate projections to station scale using spatial interpolation). Only a few studies used data from soil maps that included *AWC* as input data (i.e. Fraga et al., 2013; Moriondo et al., 2013), but often at a spatial resolution still too coarse to represent the heterogeneity within growing regions. Recently, fine scale variability within growing regions has been assessed and modelled within the ADVICLIM project but focusing only on temperature (Quénol et al., 2014; Le Roux et al., 2017).

In addition to weather conditions, the water balance of grapevines also depends on vineyard geometry (row spacing, canopy height etc.), the training system (canopy shape), soil management practices and particularly site–specific factors such as *AWC*, slope and aspect (Hofmann and Schultz, 2015). These factors describe the interaction of vineyard site microclimate with water supply and atmospheric demand (Hoppmann et al., 2017; Sturman et al., 2017). *AWC*, slope and aspect are particularly heterogeneous in regions of complex terrain resulting in variability in the supply of and demand for water. Increasing water scarcity can put economic pressure on established growing regions, because severe drought stress causes losses of grape quality and yield. Adaptation measures such as the implementation of irrigation systems are expensive and access to water in many places is restricted and difficult. Although irrigation of grapevines has been allowed since 2002 in Germany, water withdrawal rights may also need to be adapted if water is taken from groundwater or surface water bodies. Since precipitation patterns are highly variable in space and time, it is problematic for growers and stakeholders to assess future developments and to make decisions for long–term mitigation and adaptation measures. Against this background, the identification of those vineyard plots or sites within growing regions likely exposed to an increasing risk for drought stress in the future can support the decision making process.

Therefore, the main objective of the study is to quantify the likelihood of risk of future water deficit on the spatial scale of individual vineyard plots within two German grape growing regions, Rheingau and Hessische Bergstraße. The scientific process included (i) statistical downscaling of an ensemble of climate model simulated data to the scale of station data, (ii) combining information from land registers, high–resolution soil maps and digital elevation models in order to characterize vineyard landscapes and their microclimate, (iii) performing vineyard water balance simulations driven by observed and simulated weather data for all vineyard plots.

## 2 Material and Methods

### 2.1 Study area, soil and climate conditions

The risk analysis was conducted for two out of the thirteen German winegrowing regions, the Rheingau and the Hessische Bergstraße, both located in the federal state of Hesse (Fig. 1). In the Rheingau, grapevines are cultivated on an area of 3191 ha (Destatis, 2018). The Rheingau is physiographically divided into the regions of upper and lower Rheingau (Löhnertz et al., 2004). The upper Rheingau includes an area of approximately 25 km length and 3–6 km width between Wiesbaden and Rüdesheim, bounded by the Rhine river to the south and the ridge of the Taunus mountain range in the north, as well as the vineyards near Hochheim on the Main river. Grapevines are cultivated between approximately 80–280 m altitude on a gently rolling hillscape. For most of the region, the soils developed from loess or sandy loess as parent material. They are fertile and have a balanced water budget. Soil erosion, intensified by agriculture over thousands of years, filled dells and in conjunction with soil formation by a variety of basement rocks (sand, clay, marl, limestone), led to the further differentiation of soils, where the loess layers were thin. The soils of the lower Rheingau to the west of Rüdesheim are very different. The direction of the Rhine changes towards north here into the Upper Middle Rhine Valley with its steep slopes. The parent material of the soil formation consists mainly of shallow glacial solifluction layers containing a lot of basement rock (sandstone, quartzite, slate). These soils are nutrient–poor, stony and shallow and generally have a low *AWC* (Löhnertz et al., 2004; Böhm et al., 2007). The second winegrowing region of Hesse, the Hessische Bergstraße, has a cultivated area of 462 ha (Destatis, 2018). The vineyards are located on the western slopes of the Odenwald mountain range, and at the eastern edge of the Upper Rhine Plain. Soils developed from loess are also dominating here. About 60 % of the soils are deep and rich with an *AWC* exceeding 200 mm, while about 20 % of the soils have an *AWC* below 125 mm, particularly at sites where the rooting depth is limited to 60–100 cm (Löhnertz et al., 2004).

The longest running weather station at Geisenheim (since 1884 in close proximity to the University and serviced by the Deutscher Wetterdienst, DWD, German Meteorological Service) had an average growing season temperature (AGST; Apr–Oct) for the reference period 1961–1990 of 14.5 °C and 548 mm annual precipitation. Spatial variation of temperature or precipitation within both regions is relatively small. For a more recent period (2014–2018) and based on data from an array of weather stations (station specific temperature data for earlier or longer periods were only available for a limited number), AGST data in the Rheingau ranged from 15.9 °C (station Frauenstein, elevation 151 m) to 16.9 °C (stations Ehrenfels, 101 m, and Erbach, 86 m) compared to 16.3 °C for Geisenheim (Rheingau), and 17.0 °C at the station Heppenheim (119 m) in the Hessische Bergstraße (see stations in Fig. 1). Annual precipitation (based on data available from 1959–1988 for various stations) varied from 545 mm (Geisenheim) to 636 mm in the Rheingau and from 750 mm to 824 mm in the Hessische Bergstraße and is almost evenly distributed over the year. Further details about precipitation characteristics are shown in Table 1.

### 2.2 Observed and synthetic weather series

In order to run the water balance model, transient daily data for temperature, global radiation, relative humidity, wind speed and precipitation are required. Air temperature is used to model the development of grapevines and cover crops over annual

cycles, and, together with global radiation, wind speed and relative humidity, to calculate reference evapotranspiration ($ET_0$) according to Allen et al. (2005). We worked with four time series, two observed and two synthetic, which are described in more detail in the following sections.

### 2.2.1 Observed weather data

The first observed series included daily weather data from 1959–1988 (the recording ended in 1989 at some of the stations) of 10 weather stations (6 in the Rheingau and 4 in the Hessische Bergstraße) distributed across the regions (Fig. 1, Table 1) and were provided by the DWD (2018). Precipitation was recorded at all stations. At the station Bensheim (Hessische Bergstraße) temperature and relative humidity were additionally measured. The station Geisenheim (Rheingau) provided data for all five weather variables. More precisely, sunshine hours (SH) were measured here over the complete period providing a proxy for global radiation (GR). A parallel measurement period of GR and SH at Geisenheim between 1981 and 1990 was used to establish correlation coefficients between these parameters (Hofmann et al., 2014) based on the Angstroem–Prescott equation and GR was calculated accordingly. In order to be able to use time series with all five weather variables for each station in the subsequent analysis, missing temperature and relative humidity data at the stations in the Hessische Bergstraße were set equal to the measured data from Bensheim and at the stations in the Rheingau with the data measured at Geisenheim. Wind speed and GR at all stations were set equal to the data measured at Geisenheim. These data were used as model inputs for an assessment of the drought stress occurrence in the past as well as to calibrate the weather generator with respect to the observed baseline climate for all stations (see details below).

The second series included daily data from 2014–2018 and came from newly established weather stations (Fig. 1) by the University. These data were used for an assessment of observed drought stress in the recent past.

### 2.2.2 Synthetic data produced by a weather generator

Input weather series representing the baseline and future climate conditions were produced by the parametric stochastic weather generator (WG) M&Rfi, which is an improved follow–up version of Met&Roll generator (Dubrovský et al., 2000; Dubrovský et al., 2004). Met&Roll was based on the Wilks's (1992) version (adopted for use in future climate conditions) of the classical parametric generator developed by Richardson (1981). M&Rfi is a single–site multi–variate daily weather generator, in which the precipitation time series is modelled by a first–order Markov chain (occurrence of wet/dry days) and Gamma distribution (precipitation amount on wet days). The non–precipitation variables are simulated by a first–order autoregressive model whose parameters depend on wet/dry status of a given day. The M&Rfi generator has been used in many climate change impact experiments (e.g. Rötter et al., 2011; Hlavinka et al., 2015; Garofalo et al., 2019). This generator also participated in a complex validation experiment of the so–called VALUE project aiming at comparison of various downscaling approaches (Maraun et al., 2019; Gutiérrez et al., 2019; Hertig et al., 2019). Two types of synthetic time series were produced by M&Rfi (Fig. S1 in the Supplement shows a flow diagram). The first time series representing the present (baseline) climate was used to validate the generator by comparing selected weather statistics derived from synthetic vs observed weather series. The second one representing the future climate was used to assess changes of the drought stress occurrence for future climate change scenarios. In producing the first time series, WG parameters representing the statistical structure of the weather variability between 1959–1988 were derived from the observed station data (baseline climate), and then a 112–year synthetic series (1989–2100) representing the baseline climate (i.e. assuming no climate change) was produced by the WG. For the climate change scenarios (second series), we modified the WG parameters based on climate change scenarios derived from 10 future climate simulations made within the frame of the ENSEMBLES project (van der Linden and Mitchell, 2009; Table 2). Here, RCMs were used, which were driven by various Global Climate Models (GCMs) (Table 2) and run for the A1B emission scenario and approximately 25 km grid resolution. For each station and climate simulation, the data of the four nearest RCM grid boxes enclosing the weather stations were used to derive changes in WG parameters representing the RCM–based climate

change scenario for 2058–2087. In order to construct transient time series consisting of observed data from 1961–1988

followed by synthetic weather data until 2100 (assuming a smooth increase in climate change signal), the WG parameters representing a given year $Y$ were defined by modifying the WG parameters of the baseline climate with a climate change scenario, which was obtained by scaling the RCM–based scenario with a factor $k(Y, \text{ES})$, defined as

$$k(Y, \text{ES}) = \frac{T_G(Y; \text{-ES}) - T_G(1973; \text{ES})}{T_G(2073; \text{A1B}) - T_G(1975; \text{A1B})}, \qquad (1)$$

where $T_G$ is the annual global mean temperature simulated by MAGICC(v.6) (Meinshausen et al., 2011) and ES denotes a

chosen emission scenario. $T_G(1973; \text{ES})$, $T_G(2073; \text{A1B})$ and $T_G(1975; \text{A1B})$ refer to the centre years of the observed baseline (1959–1988), the RCM–future (2058–2087) and RCM–baseline (1961–1990) time slices, respectively, used to derive the WG parameters. We chose the high baseline emission scenario RCP8.5 and the medium stabilization scenario RCP4.5 (van Vuuren et al., 2011) to calculate $k(Y, \text{ES})$ (Fig. S2 in the Supplement). Therefore, the synthetic series representing the future climate were produced for RCP8.5 and RCP4.5. Results for RCP4.5 are shown in the Supplement. MAGICC is a reduced complexity

climate model, which can simulate evolution of the annual global mean temperature for a chosen emission scenario and climate sensitivity.

Climate models of the ENSEMBLES project were used instead of the successor project EURO-CORDEX (Jacob et al., 2014) for reasons of data availability at the time the study was started. Since Kotlarski et al. (2014) reported comparable biases for both projects and since it can be deduced from Feldmann et al. (2013) that the benefit from the higher spatial resolution of

EURO-CORDEX is small in the area of the study region, we concluded that the ENSEMBLES data were suitable.

The chosen RCMs were evaluated in several studies. Model errors and statistics of precipitation and temperature were analysed by Frei et al. (2003), Kjellström et al. (2010) and Suklitsch et al. (2011). Maule et al. (2013) evaluated the RCMs using drought statistics. The models showed reasonable skills in projecting weather characteristics relevant for our study.

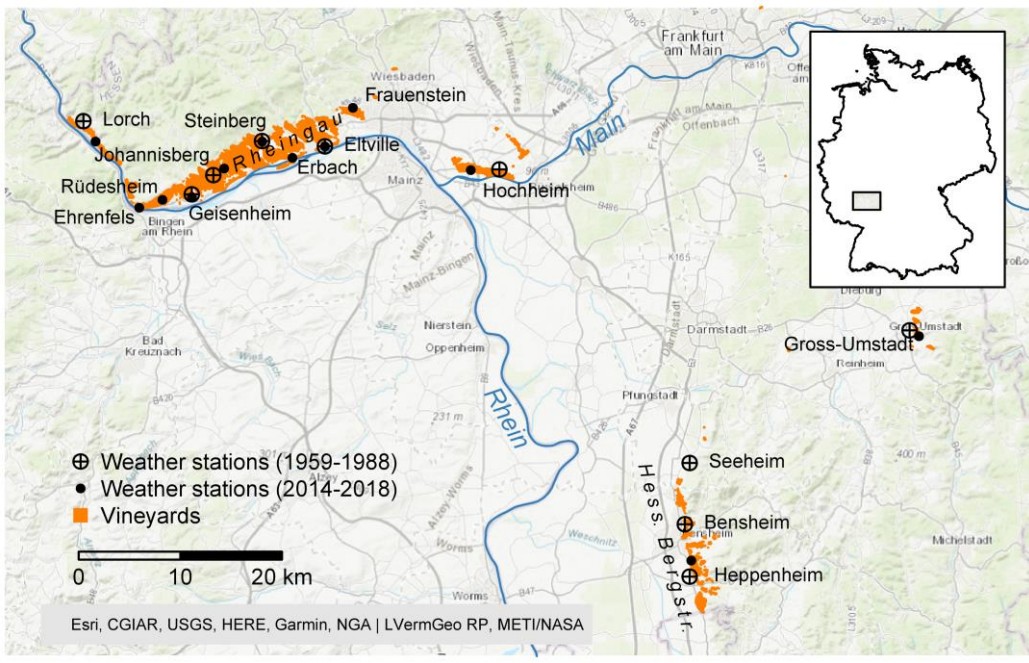


**Figure 1: Map showing the winegrowing regions Rheingau and Hessische Bergstraße and the locations of weather stations (Source of the base map (modified): Esri, 2012).**

**Table 1: Description and parameters for precipitation ($P$) of the weather stations used to calibrate the weather generator with data from 1959–1988 and to analyse the drought stress occurrence of the past. Weather data were extracted from the database of the**
**DWD (2018) and are available online at https://opendata.dwd.de.**

| Station | Stati­on-ID | Region | Ele­va- | Latitude | Longitude | Annual mean $P$ 1959– | Rainy days (days year[-1]) | $P$–intensity (mm rainy day[-1]) | Min. monthly $P$ | Max. monthly $P$ | Max. daily $P$ |
|---|---|---|---|---|---|---|---|---|---|---|---|

| | | | tion (m) | | | 1988 (mm) | | | (month; mm) | (month; mm) | 1959-1988 (mm) |
|---|---|---|---|---|---|---|---|---|---|---|---|
| Bensheim | 355 | Hess. Bergstr. | 117 | 49.6961 | 8.6267 | 824 | 172 | 4.8 | Feb; 49 | Jun; 90 | 91 |
| Heppenheim | 2138 | Hess. Bergstr. | 101 | 49.6500 | 8.6333 | 806 | 170 | 4.8 | Feb; 50 | Jun; 88 | 70 |
| Seeheim | 4646 | Hess. Bergstr. | 132 | 49.7500 | 8.6333 | 770 | 149 | 5.2 | Feb; 42 | Jul; 86 | 65 |
| Groß-Umstadt | 1815 | Hess. Bergstr. | 168 | 49.8667 | 8.9333 | 750 | 157 | 4.8 | Feb; 44 | Jun; 82 | 95 |
| Lorch | 3062 | Rheingau | 90 | 50.0508 | 7.8064 | 604 | 164 | 3.7 | Feb; 36 | May; 65 | 76 |
| Geisenheim | 1580 | Rheingau | 118 | 49.9864 | 7.9542 | 545 | 168 | 3.2 | Feb; 33 | Jul; 58 | 55 |
| Johannisberg | 1581 | Rheingau | 177 | 50.0033 | 7.9836 | 602 | 159 | 3.9 | Feb, 36 | Jul; 64 | 61 |
| Steinberg | 1213 | Rheingau | 197 | 50.0333 | 8.0500 | 636 | 169 | 3.8 | Feb; 39 | Jul; 64 | 53 |
| Eltville | 1212 | Rheingau | 96 | 50.0286 | 8.1358 | 606 | 163 | 3.7 | Feb; 35 | Jul; 65 | 67 |
| Hochheim | 2242 | Rheingau | 115 | 50.0083 | 8.3738 | 586 | 171 | 3.4 | Feb, 34 | Aug; 63 | 77 |

**Table 2: Ensemble of climate models (van der Linden and Mitchell, 2009).**

| Institute | Climate–Simulation (RCM–GCM) |
|---|---|
| C4I | RCA3–HadCM3Q16 |
| DMI | HIRHAM5–ARPEGE |
| DMI | HIRHAM5–ECHAM5 |
| DMI | HIRHAM5–BCM |
| ETHZ | CLM–HadCM3Q0 |
| KMNI | RACMO2–ECHAM5 |
| MPI–M | REMO–ECHAM5 |
| SMHI | RCA–BCM |
| SMHI | RCA–ECHAM5 |
| SMHI | RCA–HadCM3Q3 |

## 2.3 The water balance model

We used the vineyard water balance model of Hofmann et al. (2014) which was developed and validated with the general
growing and cultivation conditions of the study area presented here. This model accounts for different soil cultivation (bare soil, use of cover crops, or alternating use of both), and the impact of slope and aspect of the vineyard plots on received global radiation and $ET_0$. It includes a radiation partitioning model to separate $ET_0$ between grapevines and the soil based on Lebon et al. (2003) and accounts for different vineyard geometries. The development of the foliage of the grapevines and the cover crops is modelled based on temperature summations making it possible to run the model for multi–year applications. As heavy
precipitation events are rare in the area of study, the original model did not account for surface runoff. To include possible changes of precipitation intensity in the future, the widely used curve number (CN) method (Cronshey et al., 1986; Woodward et al., 2003) was added to the original model. This procedure was developed for small watersheds taking into account that rainfall data are often only available in the form of daily values and was tested in a vineyard soil water dynamics study (Gaudin et al., 2010). The curve numbers are available in the form of tabled values and depend on the soil type, infiltration capacity of
the soil and on the type of land use. We used CN = 86 for bare soil and CN = 58 for plant covered soils (Cronshey et al., 1986). Adjustments of the CN values depending on the antecedent moisture conditions before the wetting event were conducted as described in Maniak (2010). The impact of degree of slope on runoff was neglected, because several authors reported no clear findings (Emde, 1992, based on experiments on vineyards in the Rheingau area) or a small increase in runoff (Huang et al., 2006; Ebrahimian et al., 2012) within the range of the measurement accuracy of precipitation.
Since the individual geometry of each vineyard plot within the two regions was unknown, the calculations of radiation interception were performed for a uniform geometry representing a standard vertical shoot positioning system in Germany (Hoppmann et al., 2017). We used 2 m row spacing, a foliage width of 0.4 m, a maximum row height of 2.10 m and minimum

height of the lower end of the canopy of 0.8 m above the soil surface as base data. This is typical for the current and mid-term future situation, because more than 80 % of the new planted vineyards from 2002–2013 (> 30 % of the region) have a row spacing of 180–200 cm (data from the EU vineyard register; RPDA, 2012).

For the simulations, it was assumed that in the Rheingau, soil cultivation and cover crops alternate in every second row, and in the Hessische Bergstraße, the soil is completely covered by vegetation (except for a strip of 0.4 m under the vines). This is currently typical for both regions.

## 2.4 Soil data, spatial resolution and linking of vineyard plots to climate data

The study is based on the high spatial resolution of individual plots (Fig. S3 in the Supplement shows a flow diagram). The underlying data were provided in digital form as spatial polygons from the local authority in charge of the official EU vineyard register (RPDA, 2012). This resolution can be considered as high, with a total plot number of planted vineyards of 24858, with a mean area of 0.15 ha per plot. Plots up to 0.5 ha take up 79 % of total planted area with a maximum plot size of 4.2 ha. Each plot was linked with a digital elevation model at 1 m resolution to calculate the mean slope and aspect of the plots. The water balance model needs values for the available water capacity up to 2 m depth ($AWC_{2m}$) as the reservoir for grapevines, and 1 m depth ($AWC_{1m}$) as the reservoir for cover crops, and a value for the total evaporable water ($TEW$) of the soil surface layer, in order to calculate bare soil evaporation according to Allen (2011). The $TEW$ is defined as the difference between the water content at field capacity and half of the water content at the permanent wilting point for the upper soil layer of 10–15 cm. The $AWC_{2m}$, $AWC_{1m}$ were directly and the $TEW$ indirectly obtained from a soil database of the official state map series BFD5W (HLNUG, 2008). The data of the BFD5W are based on soil samples that were taken down to a maximum depth of 2 m at intervals of 20 m and 25 m (Böhm et al., 2007). In general, the data include the main root horizon of established vineyards (Smart et al., 2006) and take into account the lower root horizon and $AWC$ on shallower soils. Rooting systems of young vineyard (especially in the first three years) are not fully established. Those vineyards take up an area of 6–10 % and are much more vulnerable to drought stress (Fig. S4 in the Supplement). They are a special case and were not investigated in this study. To calculate the $TEW$, the BFD5W provides data on water, gravel and clay content for the plough horizon. We then used the methods described in Vorderbrügge et al. (2006) to estimate the water content at field capacity and the wilting point. The $TEW$ was determined for the upper 10 cm soil layer in this study.

For each vineyard plot, the climate data of the nearest weather station were used to calculate the water balance. For almost all vineyards (> 99 %) the distance to the next station was less than 8 km with a mean distance of 2.3 km. Figure S5 in the Supplement shows differences in observed and projected annual precipitation of the stations.

## 2.5 Assessment of drought stress

As an indicator of drought stress, we calculated the yearly sum of drought stress days during the vegetation period (1 May–30 September). A day was classified as a drought stress day, if the remaining soil water content was smaller than 15 % $AWC_{2m}$. This approach is based on the assumption, that the $AWC_{2m}$ of the soil maps corresponds to the total transpirable soil water ($TTSW$) used in earlier water balance studies (Lebon et al., 2003; Hofmann et al., 2014). The chosen threshold value of $AWC_{2m}$ corresponds to the common plant physiological threshold value for severe stress of -0.6 MPa vine predawn leaf water potential ($\psi_{pd}$) as described in Hofmann et al. (2014).

## 2.6 Spectral Analysis

The R-package multitaper (Karim et al., 2014) was used to compare power spectra of observed and synthetic time series (see the following section 3.1). This package provides also a harmonic $F$-test to assess the significance of harmonic components found in a time series.

# 3 Results

## 3.1 Validation of the weather generator

Downscaling data of climate models to station data is not trivial and all of the possible methods have pros and cons, which have to be considered in order to interpret the data and results (Maraun et al., 2010). In order to assess specific features of the used downscaling approach, we compared 30 consecutive years of the first synthetic time series (representing the baseline climate, see section 2.2.2) produced by the WG for the station Geisenheim with observational weather recorded at Geisenheim from 1959–1988. We compared the characteristics of the weather variables precipitation, global radiation and the derived $ET_0$ because of their impact on the water budget. The station Geisenheim was selected, because all required weather variables were measured continuously here. We used power spectra to compare the features of both time series regarding cyclic patterns (Fig. 2a-c), kernel density estimates to compare frequency distributions of daily data (Fig. 2d-f), monthly means to compare seasonality (Fig. 2g-i) and annual means to compare interannual variability (Fig. 2j-l). The power spectra (Fig. 2a-c) show the contributions of frequencies in the range of 1/15 year$^{-1}$ to 12 year$^{-1}$ (corresponding to cycles with periods of 15 years to 1 month). Based on the $F$-test (Karim et al., 2014) the annual cycle (at a frequency of 1 year$^{-1}$) was identified for $ET_0$ and global radiation as a significant period for both the observed and the synthetic times series. The peak at a frequency of 2 years$^{-1}$ (half year period) of the $ET_0$ power spectra (Fig. 2b) is likely a high-order harmonic of the annual cycle. Significant cycles with periods different from the annual cycle could not be identified. No clear periods were found for precipitation (Fig. 2a). The synthetic series show lower spectral contributions for periods longer than one year (frequency < 1 year$^{-1}$) and for periods in the range of 2 months to 1 month, which is related to the natural variability of the climate (like the ENSO signal or the persistence of weather patterns) which is not explicitly modelled by the WG. Frequency distributions (Fig. 2d-f) and seasonality (Fig. 2g-i) of the weather variables were well reproduced. Since the WG does not model long-term variations, interannual variability is underestimated (Fig. 2j-l).

Further comprehensive statistical validation studies were already performed in the framework of the VALUE project (Maraun et al., 2015), where the M&Rfi WG was a member of a large ensemble of statistical downscaling methods. Briefly summarized, the WG showed small biases for most of the climate characteristics studied, but underestimates were reported for precipitation variability (Gutiérrez et al., 2019), interannual variability (temperature and precipitation; Maraun et al., 2019) and long wet or dry spells (Hertig et al., 2019).

To perform an indirect validation of the WG, we compared the annual sum of drought stress days during the vegetation period (section 2.5), calculated with the water balance model for both the observed and the synthetic time series of three weather stations: Geisenheim, Hochheim (in the west and east of the Rheingau) and Bensheim (Hessische Bergstraße). In order to get a valid drought stress response to the weather data, the water balance model was parameterized for a vineyard with a comparable low $AWC_{2m}$ of 100 mm. Figure 2m-o shows histograms of the number of drought stress days of the 30-year time series for the three stations. For all stations, the frequency of years with a medium amount of drought stress days was well reproduced, but years with no drought were overestimated and years with many drought stress days were underestimated by the WG. This higher frequency of dry years in the observed data is related to long-term variations of the climate system, which, as described above, are not modelled by the WG. Based on these results we concluded that it is possible to model long-term trends regarding the development of drought stress using the climate change scenarios generated by the WG (second series, see section 2.2.2). However, it should be noted that frequencies of dry and wet years are underestimated.

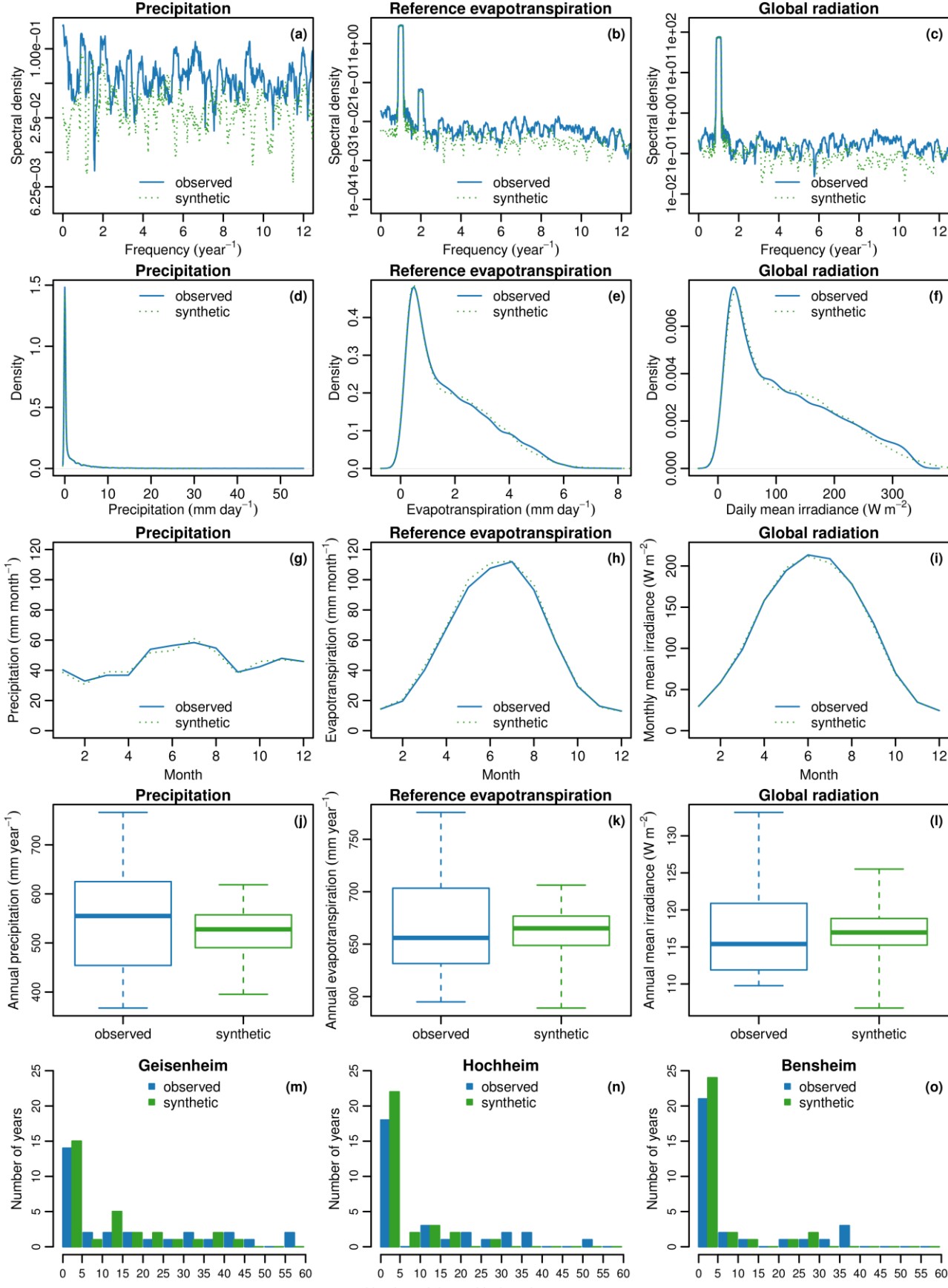

Figure 2: Comparison of 30 years of observational weather data (1959–1988, station Geisenheim, Rheingau, Germany) and 30 years of synthetic weather data produced by the weather generator calibrated with the observational weather data. Top row (a-c): Power spectra of daily data of precipitation, reference evapotranspiration and global radiation. Second row (d-f): Probability density functions of daily data computed with kernel density estimates. Third row (g-i): Monthly means. Fourth row (j-l): Box-and-whisker plots of annual values. The central box shows the interquartile range, the bold line the median; the whiskers extend to the extremes. Bottom row (m-o): Histograms of the number of drought stress days per year (n = 30 years) calculated with a water balance model for a vineyard with low available water capacity (100 mm) based on the data of Geisenheim (m) and two other weather stations of Hochheim (n) and Bensheim (o) for the same time period.

### 3.2 Water balance trends and drought stress occurrence based on observed weather data

#### 3.2.1 Water balance

Figure 3 illustrates the trend and interannual variability of the water availability expressed as climatic water balance and calculated with data of the weather station Geisenheim. The climatic water balance represents the difference between the sum of precipitations and the sum of reference evapotranspiration over a hydrological year (1 Nov–31 Oct). The presentation of Fig. 3 does not directly allow conclusions on the extent of drought stress of a certain year, which additionally depends on site factors and the plant response to limit water use. Nevertheless, the climatic water balance has decreased by about 90 mm, if

the two 30–year periods from 1959–1988 and 1989–2018 are compared. Additionally, the frequency of years with a climatic water balance lower than -200 mm has more than doubled over this period, from 8 to 18 years out of 30.

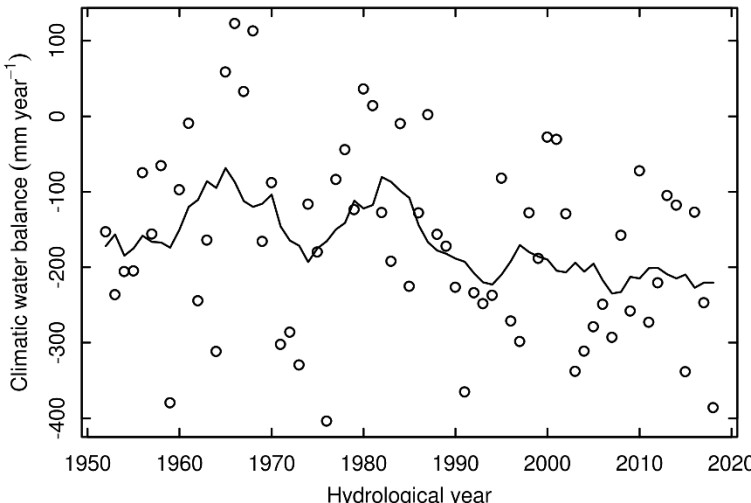

**Figure 3: Climatic water balance, expressed as the difference between the sum of precipitations and the sum of reference evapotranspiration for a hydrological year (1 Nov–31 Oct) for the station Geisenheim (Rheingau), Germany. The solid line shows**
**11–year running mean values. The decreasing trend is significant (p < 0.05, Mann–Kendall trend test; calculated with the R–package Kendall; McLeod, 2011).**

#### 3.2.2 Drought stress simulations

**Period 1959–1988:** Water balance calculations for both growing regions with the data of the observation period from 1959–1988, showed that the five driest years were 1959, 1964, 1973, 1974 and 1976. In 1959 and 1976, this was related to hot and

dry summers with many sunshine hours and high evaporative demand and in 1964, 1973 and 1974, because of extreme dry winters despite only average summers. On average, drought stress days were calculated in the range of 0–41 days per year and individual plot for the Rheingau and in the range of 0–23 days for the Hessische Bergstraße. The growing area affected by drought stress (the area with more than 10 calculated drought stress days per year on average) accounted for 6 % of the Rheingau (181 ha) and for only 1.2 % (5 ha) of the Hessische Bergstraße.

**Period 2014–2018:** Figure 4 shows the strong interannual variability of the soil water content dynamics for a typical vineyard ($AWC_{2m}$ = 110 mm, south oriented, 27.5° slope) in the steep slopes of the lower Rheingau area in the west of Rüdesheim (Fig. 1). In 2014 and 2017 moderate drought stress occurred after mid–June (after flowering) until the beginning (2014), respectively the end of July (2017), followed by moderate (2017) to wet conditions during the ripening period. The year 2018 started with a well–refilled soil profile after winter and was quite wet until mid–June, after which an extreme dry and hot period followed,

leading to a fast reduction in soil water content and severe drought stress.

The map in Fig. 5 shows the simulated spatial distribution of the sum of drought stress days for the entire Rheingau region for the year 2018 based on data of the weather station network (Fig. 1). The year 2018 had the highest sum of annual $ET_0$ (876 mm) since 1951 (first year where all weather variables to calculate $ET_0$ were recorded at the station Geisenheim). The simulations agreed with observations in the lower Rheingau (near 7.9°E and 49.98°N, Fig. 4, and between 60 to 92 days ($\psi_{pd}$)

< -0.6 MPa), where for many vineyards strong reductions in yield and restricted sugar accumulation were observed. In that particular vintage, the growing area with more than 10 calculated drought stress days was 13 % (400 ha).

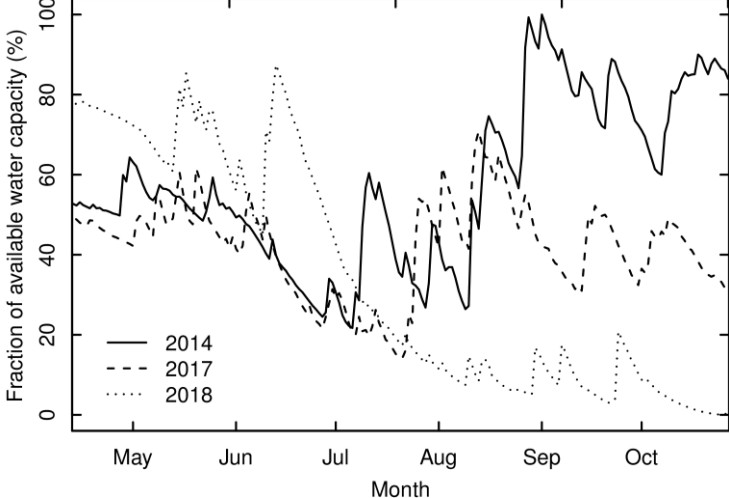

**Figure 4: Seasonal patterns of the fraction of available water capacity ($AWC_{2m}$) for a typical steep slope vineyard in the west of the Rheingau area (near station Ehrenfels, see map in Fig. 1). Simulations were conducted with a water balance model for the years**
**2014, 2017 and 2018. Parameters for the model input were: $AWC_{2m}$ = 110 mm, south oriented, 27.5° slope, 2 m row spacing, one row bare soil, one row cover crop.**

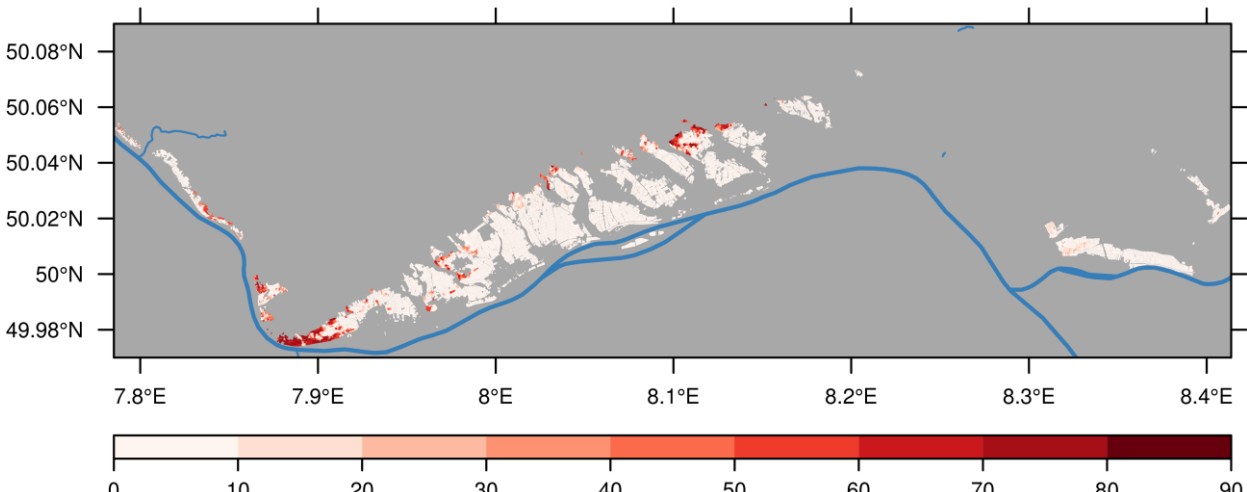

**Figure 5: Number of simulated drought stress days per vineyard plot for the winegrowing region Rheingau, Germany, during the 2018 vegetation period (1 May–30 Sep). Calculations were conducted with a water balance model based on data from weather**
**stations and a digital soil map on the assumption of alternating soil cultivation (one row bare soil, one row cover crop).**

**3.3 Water balance trends and drought stress occurrence based on climate simulations**

**3.3.1 Projected annual trends of precipitation and the climatic water balance to 2100**

Annual precipitation projected by the ensemble of climate simulations for the station Geisenheim based on the emission scenario RCP8.5 showed a high variability (Fig. 6a). The change signal (difference of mean values between the time–period
2071–2100 and the period of observed values 1961–1988) ranged from a decrease of -141 and -53 mm to an increase of +73 and +170 mm for the two most extreme simulations. Significant trends appeared at mid–century (Mann–Kendall trend test, p < 0.05, Fig. 6b). After 2073, the projected trends of seven simulations were significant, with two of them showing a decreasing trend and five an increasing trend. Compared to annual precipitation the climatic water balance per hydrological year (1 Nov– 31 Oct) decreased more strongly and ranged from -257 mm and -182 mm to +67 mm and +169 mm (Fig. 7a), because $ET_0$
increased in all simulations in a range of +3 mm to +207 mm (Fig. S6 in the Supplement). Here, the simulation projecting the strongest increase in precipitation showed the lowest increase in $ET_0$, and thus an increase in the climatic water balance of the same amount as the increase in precipitation. The two models projecting a decrease in precipitation also showed the strongest decrease in the climatic water balance (Fig. 6a and Fig. 7a), but differed in the development of individual weather variables,

especially global radiation (Fig. S7 in the Supplement). One model projected the strongest increase in $ET_0$ caused by the

strongest increase in global radiation and a strong increase in temperature, but the model with the strongest decrease in precipitation also showed the strongest decrease in global radiation and thus an increase in $ET_0$ in the medium range of the ensemble (Fig. S5–S7 in the Supplement). With one exception, the simulations that projected a moderate increase in precipitation also showed a moderate increase in $ET_0$. As a result, five simulations of the ensemble showed no trend for the climatic water balance, two showed an increasing and three showed a decreasing trend (Fig. 7). Looking at individual weather

variables, only one model showed an increase in global radiation of 10 % by 2100, while all other simulations projected a decrease of global radiation of up to -15 % by 2100 (Figure S6 in the Supplement). With regard to $ET_0$, the decrease in global radiation did not lead to a reduction and was compensated by the temperature increase (2.5 °C to 5.6 °C, Figure S7 in the Supplement). Beside the simulation projecting an increase in global radiation, the temperature increase was the only driver of the increase in $ET_0$ as the projected changes of wind speed and relative humidity were only minor (Table S3 in the Supplement).

In comparison, the ensemble results for RCP4.5 showed substantial smaller change signals for annual precipitation ranging from -63 mm to +93 mm and for the climatic water balance ranging from -149 mm to +95 mm (Fig. S9-S10 in the Supplement). Hence, the trends of the change signals of fewer simulations were significant (from approximately 2070 two increasing and two decreasing for precipitation, three decreasing and one increasing for the climatic water balance).

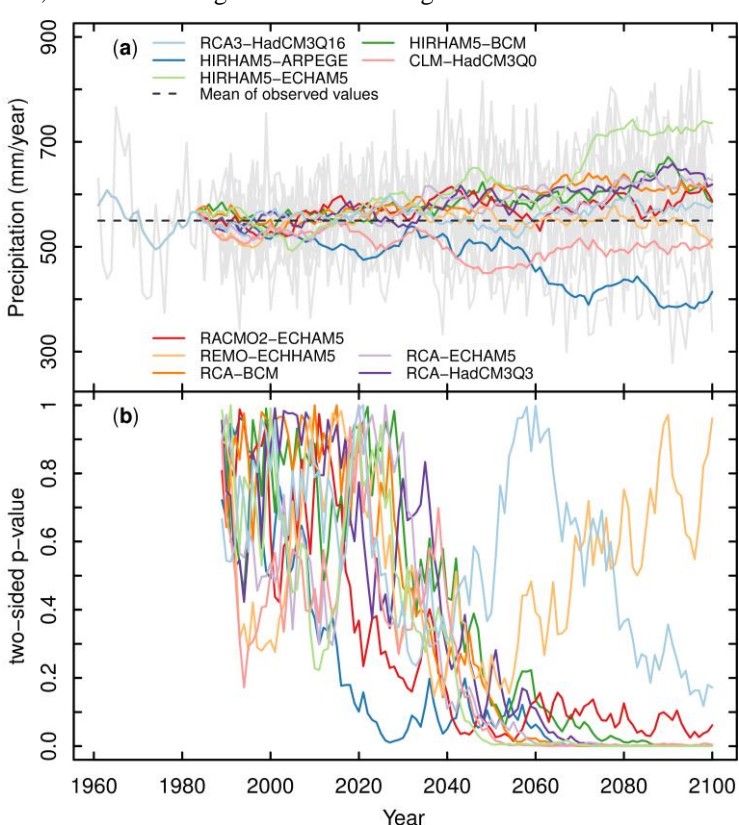

**Figure 6: (a) Annual precipitation rates of 10 climate simulations with different models for the station Geisenheim (Rheingau), Germany, for the emission scenario RCP8.5. Grey lines show the range of annual values of all models, coloured lines 11–year running means for individual model runs. The period from 1961–1988 shows observed data. (b) p–values calculated with Mann–Kendall trend test for time series of annual precipitation rates shown in (a) starting in 1961.**

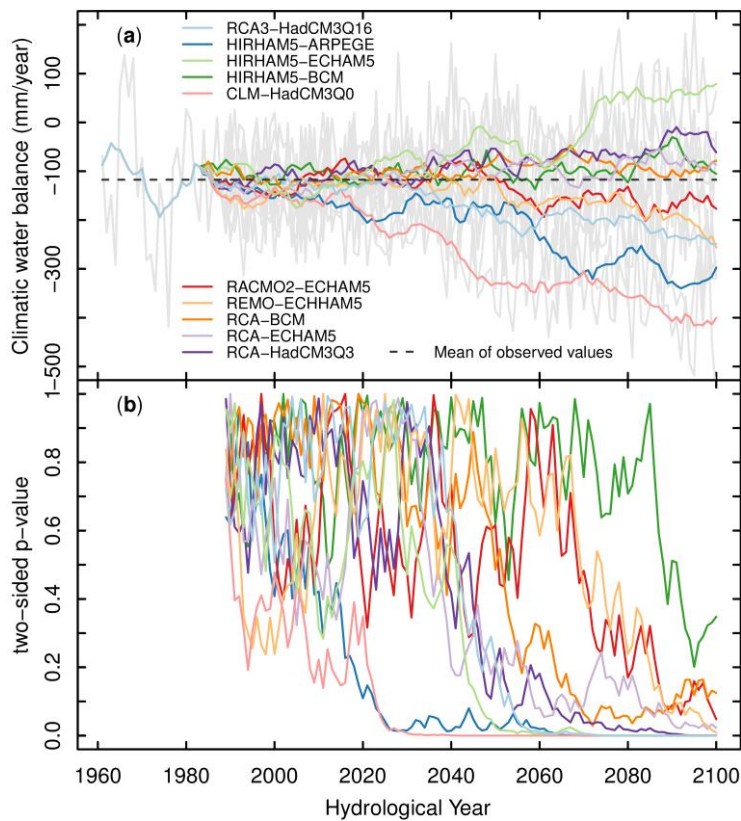

**Figure 7: (a) Climatic water balance of 10 climate simulations of different models for the station Geisenheim (Rheingau), Germany, per hydrological year (1 Nov–31 Oct) based on the emission scenario RCP8.5. Grey lines show the range of annual values of all models, coloured lines 11–year running means for individual model runs. The period from 1961–1988 shows observed data. (b) p–values calculated with Mann–Kendall trend test for time series shown in (a) starting in 1961.**

### 3.3.2 Projected seasonal trends (spring, summer, autumn, winter) of precipitation and the climatic water balance to 2100

Seasonal trends of the model ensemble for RCP8.5 are shown in Fig. 8. In part, the results of precipitation change signals (2071–2100 compared to 1961–1988, Table 3) reflected possible future seasonal shifts. The range of change signals of the transition seasons spring (March, April, May (MAM); -17 to +58 mm, Fig. 8a) and autumn (September, October, November (SON); -28 to +42 mm, Fig. 8c) is quite high, whereby in both seasons some models projected no changes of precipitation in the future up to 2100. In winter (December, January, February (DJF), Fig. 8c) all models except one (-14 mm) projected a precipitation increase (+23 to +41 mm). In summer (June, July, August (JJA), Fig. 8b), the ensemble splits up in three groups at the end of the century, one model projects an increase of precipitation (+39 mm), six models are in the range of a small decrease to no change (-22 to +1 mm) and three models project a precipitation decrease (-60 to -81 mm). In general, this indicates future seasonal shifts with an increase of precipitation in winter and a decrease of precipitation in summer.

Taking into account reference evapotranspiration by calculating the seasonal climatic water balance, the picture changed towards dryer conditions (Fig. 9, Table 3). In winter, the plus of precipitation is slightly reduced due to higher $ET_0$ (-21 to +34 mm, Fig. 9d). This is relevant in water balance calculations, because (actual) evapotranspiration is normally not reduced by dry soils due to the better water availability during these months. This also applies in parts for spring (-42 to +62 mm, Fig. 9a) and autumn (-51 to +32 mm, Fig. 9c). A clear change signal could be identified for summer, only one model projected an increase (+48 mm) all others a decrease in the range of -191 to -17 mm (Fig. 9b) due to a significant change signal for $ET_0$ in the range of -9 to +130 mm (Table 3). Climate simulations for other weather stations showed similar results (not shown).

The results for RCP4.5 showed smaller change signals for precipitation and the climatic water balance (Fig. S11-S12 and Table S4 in the Supplement). The projected increase in winter precipitation for RCP4.5 was about half as large as the increase for RCP8.5 for most simulations. Summer precipitation is also projected to decrease less in RCP4.5 compared to RCP8.5 and ranged from -43 mm to + 30 mm. No changes were projected for the climatic water balance in autumn, winter, and spring. As

$ET_0$ is projected to increase for RCP4.5 in the range of -8 mm to +72 mm in summer, compared to -9 to + 130 mm for RCP8.5, the projected decrease of the climatic water balance for summer was also less pronounced and ranged from -116 mm  to +38 mm.

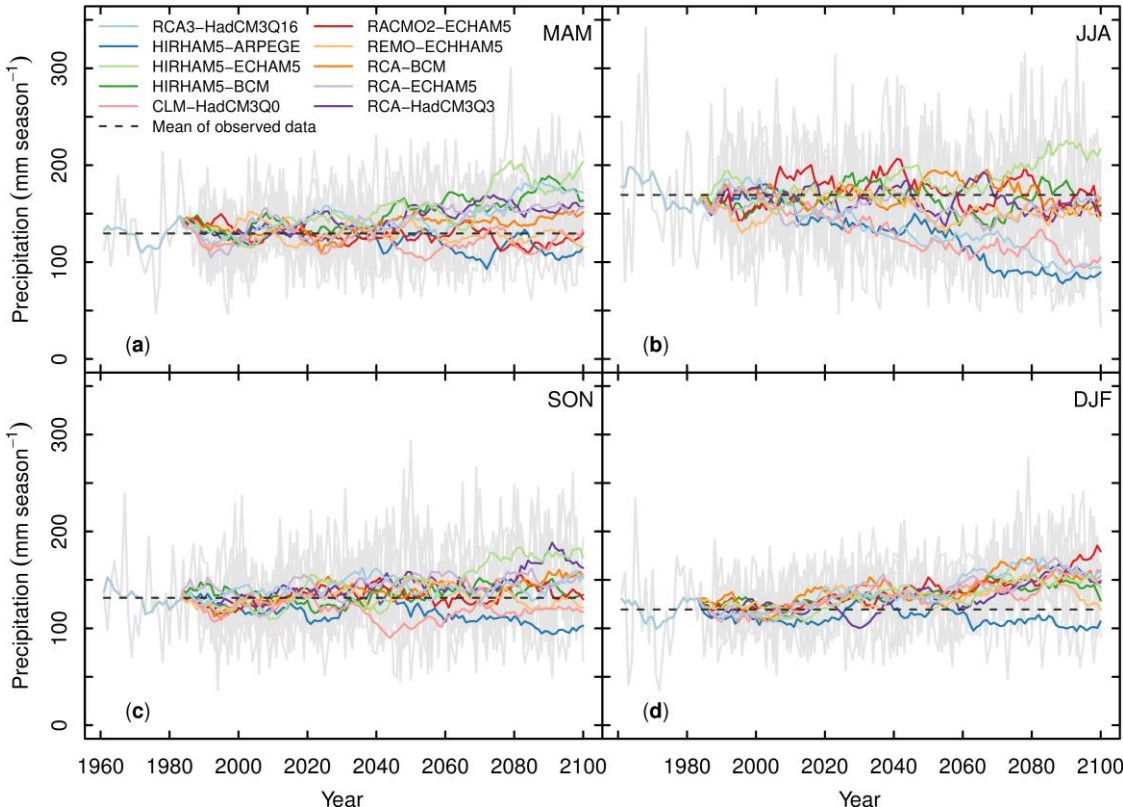

Figure 8: Seasonal precipitation simulated with 10 climate models for the station Geisenheim (Rheingau), Germany, for the emission scenario RCP8.5. Grey lines show the range of annual values of all models, coloured lines 11–year running means for individual model runs. The period from 1961–1988 represents observed data and the dashed baselines illustrate their mean values. (a) MAM, spring, March, April, May; (b) JJA, summer, June, July, August; (c) SON, autumn, September, October, November; (d) DJF, winter, December, January, February.

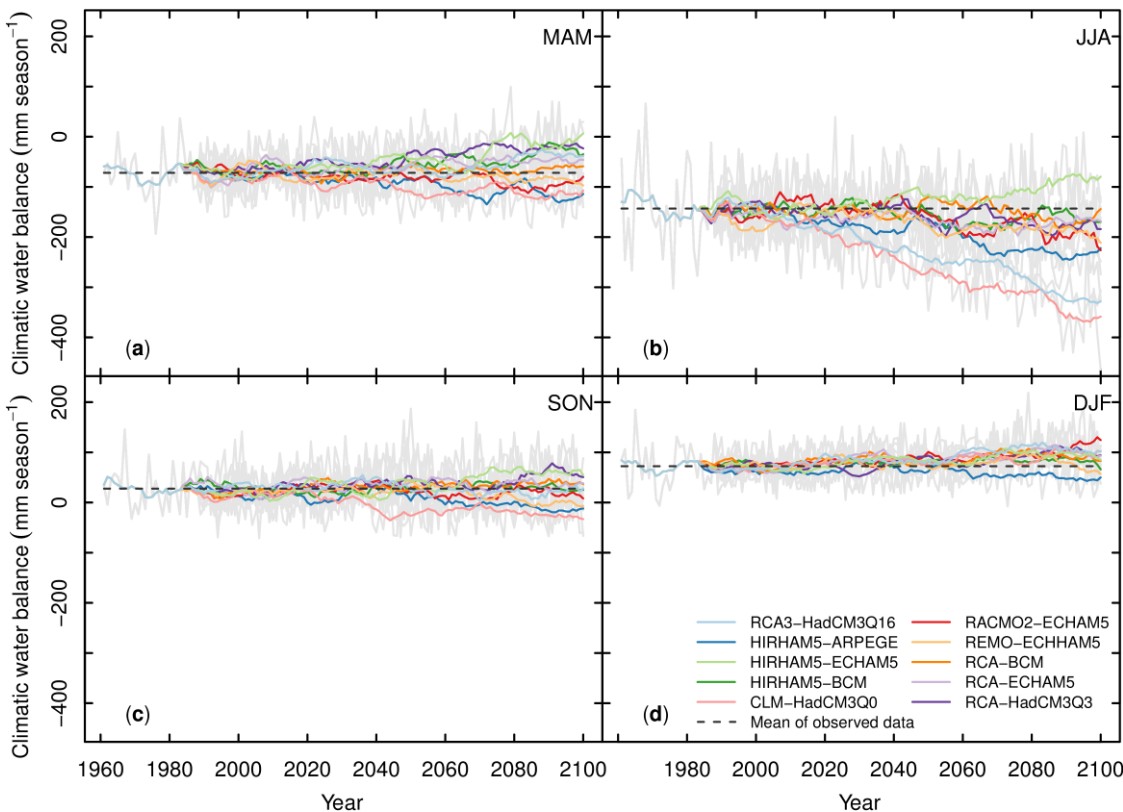

**Figure 9: Seasonal climatic water balance simulated with 10 climate models for the station Geisenheim (Rheingau), Germany, for the emission scenario RCP8.5. Grey lines show the range of annual values of all models, coloured lines 11–year running means for individual model runs. The period from 1961–1988 represents observed data and the dashed baselines illustrate their mean values. (a) MAM, spring, March, April, May; (b) JJA, summer, June, July, August; (c) SON, autumn, September, October, November; (d) DJF, winter, December, January, February.**

**Table 3: Range of change signals of 10 climate simulations with different models for the station Geisenheim (Rheingau), Germany, for the emission scenario RCP8.5. For precipitation ($P$), reference evapotranspiration ($ET_0$) and climatic water balance ($CWB$, $P$-$ET_0$) for spring (March, April, May, MAM), summer (June, July, August, JJA), autumn (September, October, November, SON) and winter (December, January, February, DJF), change signals were calculated by the difference of the individual model means between the time–period 2071–2100 and the observation period 1961–1988.**

| Season | Ensemble change signal (2071–2100 minus 1961–1988) | | |
|---|---|---|---|
| | $P$ (mm) | $ET_0$ (mm) | $CWB$ (mm) |
| Spring (MAM) | -17 to +58 | -23 to +32 | -42 to +62 |
| Summer (JJA) | -81 to +39 | -9 to +130 | -191 to +48 |
| Autumn (SON) | -28 to +42 | +6 to +36 | -51 to +32 |
| Winter (DJF) | -14 to +41 | +6 to +18 | -21 to +34 |
| Year | -141 to +170 | +3 to +207 | -260 to +166 |

### 3.3.3 Projected drought stress risk for the winegrowing regions Rheingau and Hessische Bergstraße

As most of the climate simulations for RCP8.5 showed significant annual precipitation trends in the second half of the century (Fig. 6b) and indicated changes in climatic water balance, we calculated the average number of drought stress days for the time–periods 1989–2018 and 2041–2070 for each vineyard plot and climate model. Based on this calculation, two indices were derived. The first one is describing the overall grape–growing surface area affected by drought stress, defined as the sum of the area of all individual vineyard plots with on average per time–period ten or more days of drought stress during the vegetation period. The second one is the drought stress change signal, calculated as the difference of the average number of drought stress days per vineyard plot and climate simulation between both time–periods. The calculation of the grape–growing surface area affected by drought showed that three models projected a substantial increase of this area for both regions of possibly 10 to 30 % (Rheingau), respectively 16–20 % (Hessische Bergstraße), for the future period 2041–2070. Among these three models were the two projecting a decrease in annual precipitation and the largest decrease in annual climatic water balance, described in more detail in section 3.3.1. The third model illustrates further future weather patterns, that could lead to a strong increases in drought stress. This model projected increasing precipitation in SON, DJF and MAM, but a strong decrease in precipitation in JJA and additionally a strong increase in $ET_0$ caused by the largest temperature increase of the ensemble. (Fig. S5–S6 in the Supplement). This led to a significant reduction of the climatic water balance in JJA. This indicates that presumably vineyards with low $AWC$, which may not be able to store enough of the increasing precipitation outside the summer months, are affected by a strong increase of drought stress due to the warmer and drier conditions. For both regions, the median of the climate model ensemble of the drought stress area increased slightly by 2 % and reflected projected changes in the range of no change to a small increase of the ensemble, while one model projected a decrease of 2 % for the period 2041–2070 compared to 1989–2018 (Fig. 10).

Similarly, for RCP4.5, seven models projected no or only small changes in the range of -2 %–+3 % of the drought stress area between the periods for both regions (Figure S13 in the Supplement). For the three models projecting an increase of the drought stress area for RCP8.5 and the period 2041–2070, the drought stress area for RCP4.5 is reduced, in case of the driest simulation in the Rheingau distinctly by half. It ranged from 9 %–14 % (Rheingau) and 10 %–14 % (Hessische Bergstraße) for RCP4.5 compared to 11 %–30 %, respectively 16 %–19 % for RCP8.5 simulations.

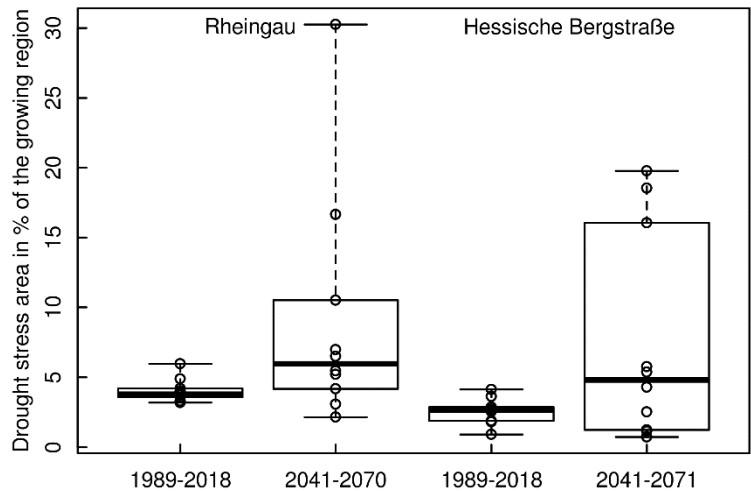

**Figure 10: Potential drought stress area of two winegrowing regions (Rheingau and Hessische Bergstraße) in Germany for two time–periods (1989–2018 and 2041–2070), calculated with a water balance model, soil maps and 10 climate simulations with different models for the emission scenario RCP8.5. A vineyard plot was allocated to the drought stress affected area, if on average 10 or more days with drought stress during the vegetation period (1 May–30 Sep) were calculated. Individual model results are shown as points in the boxplots.**

The calculation of the drought stress change signals per vineyard plot allowed the creation of maps, to illustrate spatially the impact of the projected climate trends. For RCP8.5, the maps for the "dry" and for the "wet" simulation at the extremes and the simulation close to the median of the ensemble (Fig. 10) are shown in Fig. 11 (Rheingau) and Fig. 12 (Hessische Bergstraße). In case of the dry simulation (Fig. 11a), the vineyards where drought stress already occurred in the past (in the lower Rheingau, and near Johannisberg 50.0 °N, 7.97 °E, see Fig. 1; and Martinsthal 50.05 °N, 8.12 °E, not indicated on Fig. 1) would be affected in parts (lower Rheingau) by a strong increase of drought stress. But drought stress could also increase on plots where it is at present unknown, around the two weather stations with the lowest annual rainfall, Geisenheim and Hochheim (Table 1), although many of those plots have a good *AWC* (> 175 mm; Löhnertz et al., 2004). The moderate simulation close to the median of the ensemble (Fig. 10) projected a drought stress increase up to 20 days in the Rheingau but confined to vineyard plots already affected by drought stress in 1989–2018 (Fig. 11b). In case of the "wet" simulation a moderate (but not complete) decrease of drought stress is projected, but only on plots where it occurred in the past (Fig. 11c). At the Hessische Bergstraße, the dry simulation would affect vineyards distributed over the whole region, but with a weaker change signal compared to the Rheingau (Fig. 12a). In case of the simulation close to the median, only a few plots were affected by a drought stress increase of up to 11 days (Fig. 12b). Changes for the wet simulation were negligible (Fig. 12c).

For RCP4.5 and the Rheingau (Fig. S14 in the Supplement), the "dry" and the "medium" simulation projected a much smaller increase and the "wet" simulation a stronger decrease of drought stress days compared to RCP8.5. For the dry simulation drought stress would also occur on some vineyards near Geisenheim and Hochheim with high *AWC*, but compared to RCP8.5 on an overall smaller area and less pronounced. The almost negligible increase of drought stress for the "medium" simulation would affect only sites with low *AWC*. For RCP4.5 and the Hessische Bergstraße, a smaller increase of drought stress is projected for almost the same areas compared to RCP8.5. No changes in drought stress would occur for the medium simulation and drought stress could decrease on a few plots for the "wet" simulation.

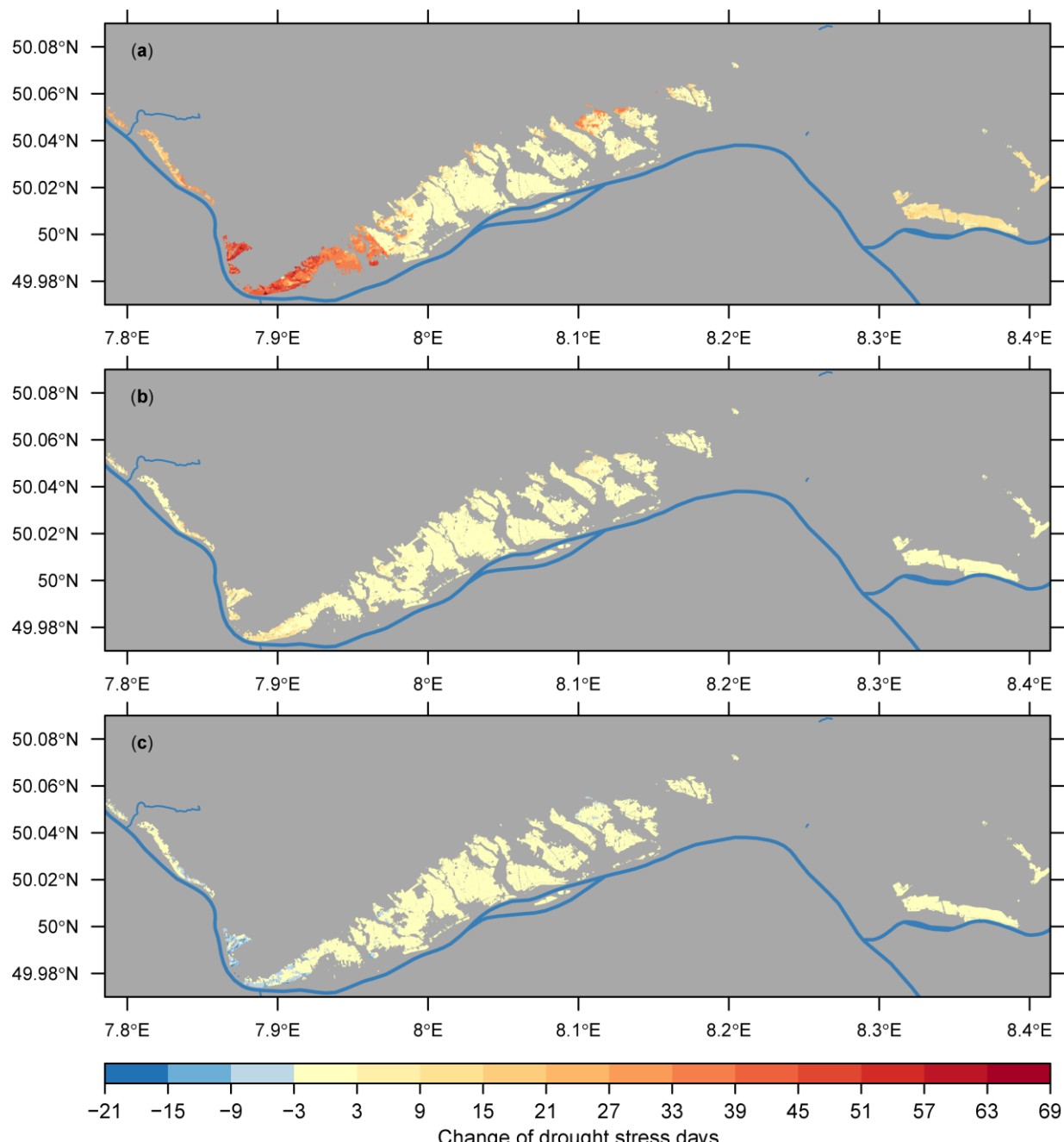

**Figure 11: Projected change of the occurrence of drought stress days for the growing region Rheingau (Germany), for the emission scenario RCP8.5, calculated with a water balance model on the assumption of alternating soil cultivation (one row bare soil, one row cover crop). The maps show the difference between the number of the mean drought stress days per vegetation period (1 May–30 Sep) for the periods 2041–2070 minus 1989–2018 at the spatial scale of the individual vineyard plots. (a) Results of the climate simulation calculating the strongest increase, (b) the simulation close to the ensemble median, and (c) the simulation projecting the strongest decrease of the drought stress area of an ensemble of 10 climate models.**

475

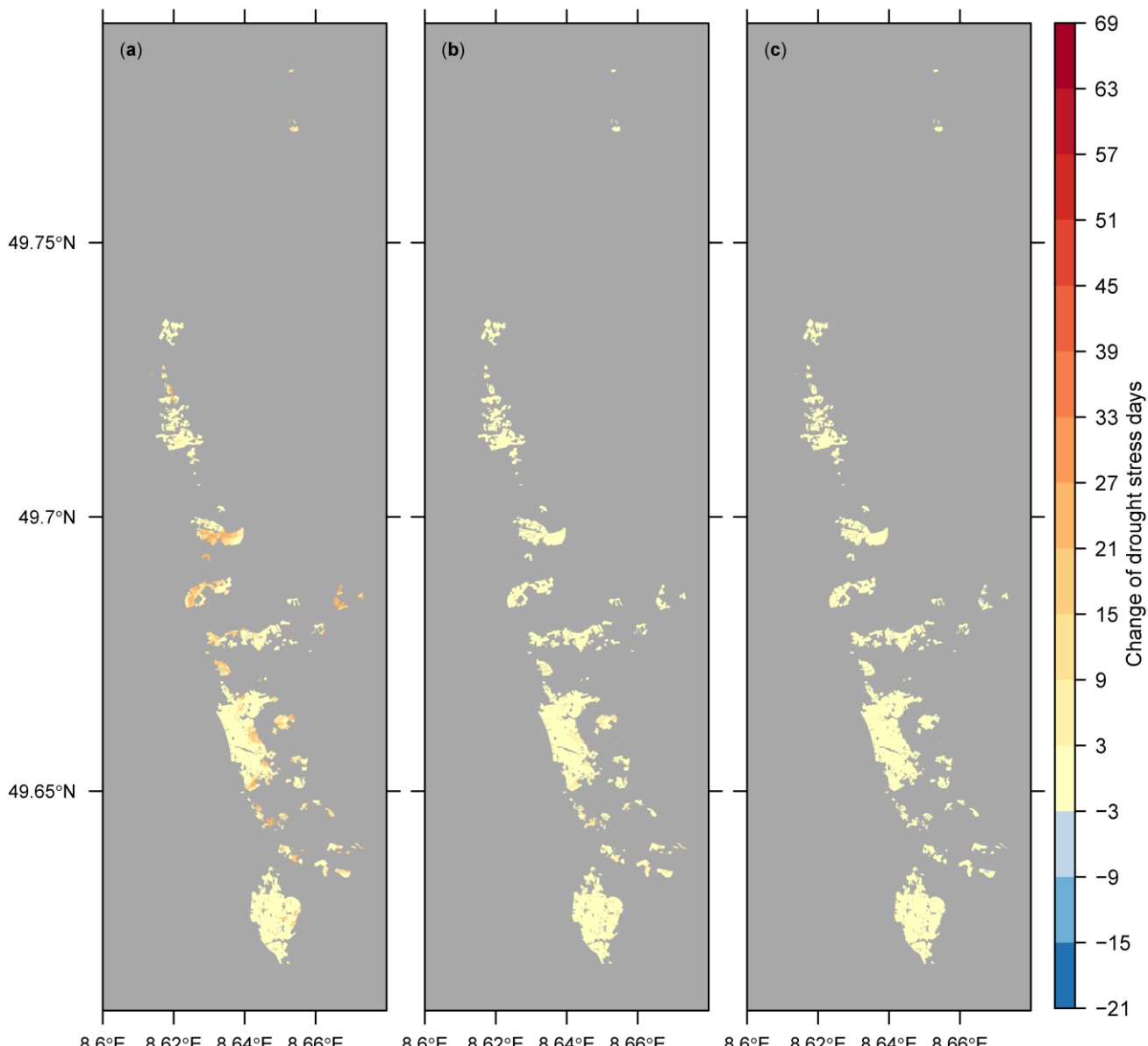

**Figure 12: Projected change of the occurrence of drought stress days for the growing region Hessische Bergstraße (Germany), for the emission scenario RCP8.5, calculated with a water balance model on the assumption of cover crop use in every row. The maps show the difference between the number of the mean drought stress days per vegetation period (1 May–30 Sep) for the periods 2041–2070 minus 1989–2018 at the spatial scale of the individual vineyard plots. (a) Results of the climate simulation calculating the strongest increase, (b) the simulation close to the ensemble median, and (c) the simulation projecting the strongest decrease of the drought stress area of an ensemble of 10 climate models.**

## 4 Discussion

### 4.1 Global and regionals aspects of the uncertainty of the projections

Climate projections and impact analyses are subject to a number of uncertainties. In the understanding of climate change, these uncertainties are in general related to the uncertain future external forcing by greenhouse gas emissions, the impact of external forcing factors on climate and the degree of natural variability of the climate system (Kjellström et al., 2011). In impact-analyses, methodical imperfections of the impact models result in further uncertainties. This study looked on a comparably small region, thus the ability of the RCMs to reproduce spatial weather patterns is one additional source of uncertainty. The water balance model itself or previous versions have been validated with field observations on different vineyard plots of the current study area as well as other regions and in different climates (Lebon et al., 2003; Pellegrino et al., 2006). Yet, on a regional scale, it requires high quality soil data, which have a strong influence on the result of the calculations as a possible source of error. The soil data go back mainly to soil mappings conducted from 1947–1958 (Böhm et al., 2007). Since then,

based on land consolidation projects, and individual interventions in parts of the complete landscape, some attributes might have changed in local spots, but in general, the soil maps are still describing the current situation quite well as demonstrated in a follow–up study (Zimmer, 1999).

To capture the magnitude of uncertainties related to possible future climate evolution for the selected emission scenario, we used climate projections for the period 2058–2087 simulated by ten climate models of the ENSEMBLES project. These data were used to derive the climate change scenario, which was further scaled by smoothly increasing change in global mean temperature (as projected by the MAGICC model for the selected RCP8.5 emission scenario) and used to modify the weather generator parameters, in order to produce transient time series for several weather stations. The simulations showed a high range of the future precipitation change at the end of the century. This range is comparable with the results of the REKLIES–DE project (±20 % for annual precipitation, region Germany and drainage basins of large rivers, 2070–2099 compared to 1971–2000), calculated with 37 climate simulations including the EURO–CORDEX data (Hübener et al., 2017; Bülow et al., 2019). Additionally, similar seasonal shifts (increase of winter and decrease of summer precipitations) were reported in this study. This range could be reduced if the extreme models at the upper or lower edge would be excluded, but since to the knowledge of the authors no severe shortcomings of the models have been reported, this would exclude possible future climate realisations. Furthermore, climate models cannot be considered as fully independent from each other (Kreienkamp et al., 2012; Flato et al., 2013), which rules out the conclusion that if the majority of results from model runs point into one direction, that this would automatically mean a higher probability for this climate realisation. However, diverging databases bear uncertainties in risk assessment and decision support processes. Noteworthy, the projected range for precipitation changes for the mitigation scenario RCP2.6 is less than half of the range for RCP8.5 (Hübener et al., 2017). A similar reduction in range was found in our results for changes in precipitation (Fig. 6a, Fig. S9a in the Supplement), the climatic water balance (Fig. 7a, Fig. S10a in the Supplement), reference evapotranspiration, global radiation, and temperature (Fig. S6–S8 in the Supplement).

One water budget simulation driven by the climate models predicted that drought stress would be less problematic in the future. This would not be expected from observations in the recent past, where drought stress occurrence has increased. The decrease in the climatic water balance (Fig. 3) is related to an increase in $ET_0$, because for the Rheingau region (station Geisenheim) no seasonal trend in precipitation rates is noticeable for the past. The observed increase of $ET_0$ is primarily a combined effect of an increase in global radiation and temperature (Table S1-S2 in the Supplement). Changes in wind velocity, as observed in other regions on the globe (Schultz, 2017), can be excluded (Table S1 in the Supplement). Relative humidity has also only marginally changed in Geisenheim, and likewise remains almost constant in the climate projections (Table S1-S3 in the Supplement). The observation that relative humidity changes are comparably small in a changing climate is well known in climate science. Manabe and Wetherald (1967) were the first to acknowledge this in their seminal paper, where they showed an almost constant relative humidity across different seasons, and then concluded that relative humidity would remain roughly constant also in a warming climate. State-of-the-art projections give still qualitatively similar results, all determined by the interplay of the law of Clausius–Clapeyron, circulation, and differential warming: over the oceans, moisture supply is essentially limited, such that evaporation increases with higher temperatures keeping relative humidity unchanged. Over the continents, relative humidity decreases slightly because the increasing moisture transport from the oceans cannot fully make up the drying of the atmosphere caused by the stronger warming of the land surface compared to the oceans. As saturation vapour pressure increases with temperature, vapour pressure deficit, a key control of evapotranspiration (Monteith and Unsworth, 2013), increases also with temperature if relative humidity remains constant (Table S1-S3 in the Supplement). Regarding global radiation, the weather recordings of the Geisenheim station clearly show the effect of global dimming (after World War II to the 1980ies) and brightening (since then) periods (Wild, 2009, 2012; Hofmann and Schultz, 2010) related to a period of strong pollution (dimming) and cleaning of the atmosphere (brightening) and observed in many places on Earth. This is reflected in an increase in mean global radiation from 116 $Wm^{-2}$ for the period from 1961–1990 to 130 $Wm^{-2}$ for the period from 1991–2020 (station Geisenheim, Table S1 in the Supplement). The strong increase in global radiation, also in

comparison with other regions of the world (Wild, 2012; possibly due to a strong decrease of sulphur dioxide emissions in Germany of 95,2 % from 1990–2019; Umweltbundesamt, 2021), probably caused a more rapid warming because the vanishing dimming no longer masked the increase of atmospheric downwelling thermal radiation caused by the greenhouse effect (Wild, 2009, 2012). This rapid warming could also be the reason why the observed increases in reference evapotranspiration, global

radiation and temperature at Geisenheim (1991–2020 versus 1961-1990) exceed the projections of the most extreme simulations for RCP8.5, which began in 1989 (Fig. S6-S9 and Table S1-S2 in the Supplement). However, the conclusion that the observed climate developments might follow the RCP8.5 pathway to the end of the century, is probably wrong following Hausfather and Peters (2020), because this emission scenario assumes increasingly unlikely high coal use. Hausfather and Peters (2020) consider a warming of 3 °C above pre-industrial levels at the end of the century, which would correspond to the

warm simulations for RCP4.5 or the cold simulations for RCP8.5 in our study. In retrospect of the observations, the most intensive droughts for the two growing regions in the years 2003 and 2018 were related to heat waves with high $ET_0$ values. Heat wave frequencies on a global scale have increased in the past (Schär et al., 2004; Kirtman et al., 2013) and are predicted to increase further in the near future, irrespective of the emission scenario (Coumou and Robinson, 2013). A study of Kornhuber et al. (2019) found that the weather extremes of the early summer 2018, where heat and rainfall extremes were

recorded in the mid–latitudes of the northern hemisphere, were connected with a persistent wave pattern in the Jetstream, which was also observed during the European heat waves of 2003, 2006 and 2015. In addition, the number of such wave patterns has increased significantly during the last two decades. Since our approach of using a weather generator had the shortcoming of reduced interannual variability, it is likely that frequencies of such extreme years are underestimated (Fig. 2j–o). As expected, the reduced interannual variability compared to station data was also found in grid box data of RCMs of the

region, because these data represent spatial means. Fraga et al. (2013) reported an increase of interannual variability of the temperature based Huglin–Index and the precipitation/evapotranspiration based dryness–index for many parts of Europe including the study region, by comparing the period from 2041–2070 with 1961–2000, calculated with 16 climate simulations from the ENSEMBLES project. On the other hand, the frequency of such extreme years is the main cause for growers to think about cost intensive adaptation measures like irrigation (Santos et al., 2020). The impact analysis for perennial crops, not only

grapevines, could profit enormously from climate simulations with the feature of well–reproduced interannual variability.

Despite of the reduced interannual variability, the climate projections showed seasonal shifts. The impact of seasonality of precipitation on grape quality is not fully understood (Sadras et al., 2012b). Dry conditions during the ripening period and harvest are in general positive for fruit quality and health, but severe drought stress can lead to a cessation of sugar accumulation, as observed in specific plots of the study area during the 2018 and 2019 vintages. Seasonal shifts of precipitation

could reduce the impact of dry spells on plots with sufficient capacity to store available water, by enhanced refilling in winter. Trömel and Schönwiese (2007) reported that the trends for the probability for observed monthly extreme precipitation in Germany varied substantial on a spatial scale and also projected near future changes of extreme precipitation showed heterogeneous spatial change patterns in summer (Feldmann et al., 2013). The performance of many downscaling and bias correction methods to represent temporal aspects of the climate has become only recently a topic of research (Maraun et al.,

2019).

**4.2. Impact Model shortcomings with respect to projected atmospheric $CO_2$ concentrations**

The water balance model currently does not account for the impact of increasing $CO_2$ on stomatal conductance ($g_s$) and transpiration. Xu et al. (2016) reported that the stomatal response to elevated $CO_2$ depended greatly on environmental variables and species and referred to studies where double ambient $CO_2$ decreased $g_s$ by 40–50 %. A general survey of the response of

stomatal aperture to an increase to 560 µmol mol$^{-1}$ in $CO_2$–concentration (from 380 µmol mol$^{-1}$; Ainsworth and Rogers, 2007) across a variety of plant species showed an approximate reduction of about 20 %. Experiments of field grown grapevines under elevated $CO_2$ showed no uniform results and ranged from an observed decrease of stomatal conductance (Everard et al., 2017)

to no significant changes (Bindi et al., 2001; Moutinho–Pereira et al., 2009) to even an increase (Wohlfahrt et al., 2018). A simple, but physically based approach to assess the impact of reduced $g_s$ on $ET_0$ is provided by the equations of Allen et al. (1998). In the Penman–Monteith equation, the bulk surface resistance for water transport is the variable depending on stomatal conductance/resistance (Lovelli et al., 2010). Applied to the weather data of the year 2017 of Geisenheim, a reduction of stomatal conductance of 20 % would lead to a reduction of 3 % of the annual sum of $ET_0$ (from 723 to 702 mm year$^{-1}$). Therefore, in the assessment of drought, the possible reduction of transpiration caused by elevated $CO_2$ is likely not the key point but there is currently a lack of knowledge about the impact of elevated $CO_2$ on the physiology of grapevines in combination with drought stress under field conditions. Additionally, depending on the grapevine cultivar, the responses to water deficit can be quite diverse (Schultz, 1996, 2003; Costa et al., 2012; Bota et al., 2016).

**4.3 Possible impacts on grape quality and cultivation caused by moderate drought stress scenarios**

The models close to the median showed for both emission scenarios a small increase of the number of drought stress days in the range of 5–20 days for vineyards of the lower Rheingau and small parts of the upper Rheingau, in general on plots where drought stress occurred already in the recent past. However, the affected area for RCP4.5 is only about half as large as the area for RCP8.5. From these simulations, some sub–regions with an increased future risk for drought stress could be identified. For already irrigated plots, the scenario outcomes mean that growers would have to irrigate between one to three times more per season on average. The used threshold value, to classify a day as a drought stress day ($\psi_{pd}$ < -0.6 MPA) represents relative severe drought stress with a strong decrease of assimilation rate (Schultz and Lebon, 2005) and cessation of vegetative growth (Van Leeuwen and Destrac–Irvine, 2017). The viticultural impact of drought stress depends also on the phenological stage when it occurs and the duration of such events. Before flowering (beginning of June –mid June), even moderate drought stress (-0.6 MPA < $\psi_{pd}$ < -0.2 MPa) is possibly negative, because it can reduce cluster size and berry numbers (Keller, 2005). Matthews et al. (1987) reported that early drought stress (before fruit softening, i.e. about beginning of August to mid-August in the Rheingau area) had a stronger impact on yield than late drought stress. Early drought stress also has a stronger impact on the final berry size (Ojeda et al., 2001). In the context of the majority of models predicting a decrease in climatic water balance in JJA (Fig. 9b) this would indicate a likely future yield effect. Impacts on quality components aside of primary compounds like sugar and acids are much more difficult to predict, vary between white and red varieties (Sadras et al., 2012b; Savoi et al., 2016) and depend on complex interactions with many environmental factors difficult to completely assess for in climate change studies (Van Leeuwen and Destrac–Irvine , 2017; Santos et al., 2020).

**4.4 Adaptation measures with respect to the local environment**

The simulations showed a widespread array of possible changes making it difficult to generalize adaptation strategies. Both viticultural regions are located in areas where nitrate leaching to the groundwater is a severe environmental issue (Löhnertz et al., 2004). This threat would certainly be enhanced in the future because of higher mineralization rates, caused by increasing temperature (both air and soil) and rainfall in winter (Table 3). The use of cover crops or natural vegetation to cover the soil on the complete vineyard surface area during the winter months is the most important measure to counteract this development (Berthold et al., 2016). Similarly, these measures and possibly reduced tillage are also important for the summer months to protect against leaching and erosion. Cover crops also reduce surface runoff and increase infiltration, but compete with the grapevines for nutrients and water. On steep slopes with shallow soils, grapevine roots and cover crops share much of the same soil reservoir. Consequently, tillage in spring and cover crops in alternate rows has become a standard praxis, balancing the advantages and disadvantages of cover crop use. Wide row spacing could reduce the water use due to the lower planting density but this would increase the risk of erosion in the cultivated rows. The possibilities to influence the water balance by canopy management are therefore limited in these situations and need also to be considered in the context with the cost disadvantages

of steep slope viticulture (Strub and Loose, 2021). A further interesting long-term viticultural adaptation strategy is the use of rootstocks with enhanced drought tolerance (Ollat et al., 2016).

On the other hand, following the climate projections, irrigation should be possible against the background of the projected shifts from summer to winter precipitation amounts and increasing annual precipitation. This generally offers the opportunity to withdraw and store water from surface water bodies during periods with high flow rates, as potential conflicts with the use of drinking water, which is usually withdrawn from groundwater bodies, could be avoided. Expanded use of bank filtration could also help to avoid future resource conflicts. The construction of such extensive infrastructure measures requires an
interplay of all actors involved.

Due to increased temperature combined with relatively unchanged but still highly variable precipitation patterns (Fig. 8c), the increased occurrence of warm and wet conditions during the ripening period (September, October) has increased the risk for rot (Schultz and Hofmann, 2015). A similar climatic trend regarding the decoupling of the relationships between temperature, drought, and early wine grape harvests was reported by Cook and Wolkovich (2016) for France and Switzerland. These types
of non-stationarities are reflected in more or less new environmental conditions and weather patterns, which are a challenge for cultivation. Apart from the water balance, these challenges in the Rheingau (like in other regions) primarily span the management of vigour, yield, grape maturity and disease management, against a background of a high terrain complexity and natural climate variability (Neethling et al., 2019). The need to assess and apply adaption measures at a regional level down to individual plots, is also evident from our study. For future impact research studies, it could be beneficial to apply regional
convection-permitting climate modelling (grid spacing < 4 km), as this approach may provide the necessary climate data for impact modelling at local level (Prein et al., 2015). This approach could also make it possible to analyse risks caused by short term extremes like hail storms, flash floods or erosion together with long term changes, because this different types of risks are finally assessed jointly in climate adaptation projects. In this respect, the application of process-based climate model evaluations taking into account synoptic weather types (Maraun et al., 2021) should be considered in future impact studies.
On the other hand, regional or local climate modelling could be improved by integrating the water balance of winegrowing regions as a land use type (Tölle et al., 2014, Hartmann et al., 2020).

**5 Conclusions**

Based on an ensemble of climate model simulations, a water balance model, a digital soil map, an elevation model and a land register, our study provides a risk assessment with respect to the future occurrence of drought stress, applied to individual
vineyard plots of the winegrowing regions Rheingau and Hessische Bergstraße. The results ranged from a small decrease (one simulation) to a moderate increase of drought stress (median of the ensemble), predominantly on plots already temporarily affected by water deficit, up to a drought stress occurrence touching 20–30 % of the growing regions. As drought stress is already currently observed in steep slope vineyards with shallow soils, these sub–regions were identified as future risk areas by most of the simulations. The results illustrate the large heterogeneity of the water supply within growing regions and
between neighbouring vineyards and the need to improve high resolution modelling approaches. Mid– and long–term adaptation measures need to respect local conditions and will necessitate individual, precision–farming–like application of cultivation practices. In combination with weather station networks delivering real time data, the presented framework may also serve as a decision support tool to growers and consultants in the future.

**Data availability.** Observed weather data of the DWD can be found at https://opendata.dwd.de/ and weather data of the Hochschule Geisenheim University at http://rebschutz.hs-geisenheim.de/wetterstationen/tagesauswertung.php. Outputs from the weather generator simulations are available on request to the corresponding author.

**Author Contributions.** MH and HS developed the concept and research goals, DM, MD, CV, and MH designed the
methodology, CV downloaded and processed the future climate change scenarios, MD programmed and calibrated the weather
generator, MH ran the water balance simulations, prepared the original draft and produced all figures, all authors contributed
to writing, review and editing.

**Competing interests.** The authors declare that they have no conflict of interest.

**Acknowledgments.** This research was funded by the Hessian Agency for Nature Conservation, Environment and Geology
(HLNUG) as a part of the INKLIM–A project. We thank Klaus Friedrich, Matthias Schmanke (HLNUG) and Christoph Presser
(Regierungspräsidium Darmstadt, RPD) for combining the comprehensive databases of vineyard plots, elevation and soil
maps. We thank Heike Hübener (HLNUG) for fruitful discussions about how to perform climate change impact studies on
individual vineyards. We also thank three referees for their careful review and helpful comments. We acknowledge support
by the Open Access Publishing Fund of Geisenheim University.

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
