# Peer review of "Downscaling of climate change scenarios for a high resolution, site– specific assessment of drought stress risk for two viticultural regions with heterogeneous landscapes"

_Earth System Dynamics, 2021_

## Author Comment (AC1)

**Response to Referee #1**

The authors thank the referee for the very detailed and constructive review and the time spent to analyse the manuscript. The review illustrates in many parts how our study should ideally have been conducted. Concerning the major points raised in relation to the used climate models we are still convinced, that the models used are suitable for our study despite the fact that there are substantial dynamics in this field and more recent models are available. Following the suggestions of the referee, our major revisions would include describing the downscaling process of the climate data to the vineyard scale including a schematic flow diagram, improving the statistical analysis of the validation of the weather generator including more weather stations, and including a further emission scenario, which will certainly improve the results. Our answers to the specific comments are given within the text in blue font below.

1. **General comments**

The rationale of the paper is based on the need to improve knowledge of local climate and soil moisture in order for vineyard regions to respond appropriately to climate change. The paper emphasises the need to downscale from global climate model predictions using regional climate models in order to assess effects on the water budget of grapevines at vineyard scale, so is in line with contemporary research trends. A weather generator and a water balance model are also used, accounting for local variations in soil characteristics, the complexity of the terrain, and crop management practices, in order to provide predictions of drought risk at vineyard scale.

The overall aim of the paper is therefore in line with objectives of international research into the application of climate models to assess impacts of and develop appropriate responses to climate change by downscaling model projections of future climate to vineyard scale. Although the general approach is fine, there are several areas of weakness in the paper, as discussed in the following section. In particular, some aspects of the methodology used in the paper seem to be rather dated and not clearly described. In particular, I would have expected that more recent climate models would have been used, given the rapid climate model development that has taken place over the past decade.

2. **Specific comments**

2.1 Soil moisture versus temperature

Line 31-32: I disagree with the general statement that 'Within the existing production areas, water shortage is probably the most dominant environmental constraint (Williams and Matthews, 1990) ….', which the authors appear to suggest applies globally. In many parts of the world, it is clear that temperature has a greater impact on grape production and wine quality, especially in 'New World' regions where irrigation is a standard practice.

**Response:** We agree that temperature has a greater impact on grape production and wine quality than water availability. Nevertheless, water is an important limiting factor not only under irrigated conditions. The statement "within the existing production areas" refers to areas, where temperature conditions are not a limiting factor for the cultivation of grapevines. The need to irrigate in 'New World' regions is an example that water shortage is an environmentally limiting factor within those regions. To avoid misunderstandings, we suggest

to write: 'Within the existing production areas, where temperature conditions are in general favourable for cultivation, water shortage is probably the most dominant environmental constraint (Williams and Matthews, 1990)… '

2.2 Dated climate models

The climate models used in this research appear to be quite old and outdated (van der Linden and Mitchell, 2009) given the rapid developments in model design and downscaling techniques over the past decade. Even the web link for the ENSEMBLES Final Report states 'This object has been archived because its content is outdated.' (https://climate-adapt.eea.europa.eu/metadata/publications/ensembles-final-report). It is therefore unclear why more recent climate models from the CMIP5 and CMIP6 evaluations, or the available EURO-CORDEX model data are not used in this work. Recent publications referenced in this paper (e.g. Gutiérrez et al. 2019) appear to suggest that EURO-CORDEX is the preferred model framework for contemporary research, and there are many publications over the past decade that have been based on CORDEX climate model data.

**Response:** We agree that using a regional climate model ensemble from the ENSEMBLES project raises the question why data from EURO-CORDEX (the successor of ENSEMBLES) were not used in this study. When we started with the project (in 2015), all nodes of the Earth System Grid Federation (ESGF) to download the EURO-CORDEX data were out of service for several months without information of a date of return. The evaluation of the performance of the climate models against observational data of Kotlarski et al. (2014) showed that the improvements of EURO-CORDEX compared to ENSEMBLES were not very significant (a detailed comparison is described in section 4.6 of this paper). Kotlarski et al. reported comparable bias ranges for EURO-CORDEX and the ENSEMBLES simulations. Since we used a weather generator in our study, we needed only the change signals of the climate simulations. High-resolution RCM ensemble simulations of Feldmann et al. (2013) showed that the relative change of mean precipitation is quite uniform in the study region (with the limitation that Feldmann et al. focused on the near future 2011-2040), but suggesting that also the higher spatial resolution of EURO-CORDEX (12 km vs. 25 km) is of limited added value for our study. Overall, we concluded that the use the ENSEMBLES instead of EURO-CORDEX would not have a significant effect on the results, and, because we also had to start with the work, we decided to use the ENSEMBLES simulations. We suggest to include parts of the argumentation above in the paper to make it clear to the reader why the used models have their value.

Feldmann, H., Schädler, G., Panitz, H.-J., and Kottmeier, C.: Near future changes of extreme precipitation over complex terrain in Central Europe derived from high resolution RCM ensemble simulations, International Journal of Climatology, 33, 1964-1977, 10.1002/joc.3564, 2013.

Kotlarski, S., Keuler, K., Christensen, O. B., Colette, A., Déqué, M., Gobiet, A., Goergen, K., Jacob, D., Lüthi, D., van Meijgaard, E., Nikulin, G., Schär, C., Teichmann, C., Vautard, R., Warrach-Sagi, K., and Wulfmeyer, V.: Regional climate modeling on European scales: a joint standard evaluation of the EURO-CORDEX RCM ensemble, Geosci. Model Dev., 7, 1297-1333, 10.5194/gmd-7-1297-2014, 2014.

The dated nature of the climate modelling component of this work is also evident by the reference in Section 2.2.2 to application of the 10 selected models to the old A1B emission

scenario (Line 152), which was developed over 20 years ago and has since been replaced by RCP scenarios (about ten years ago) and more recently by SSP scenarios (Tebaldi et al. 2021 – see below).

Tebaldi, C., Debeire, K., Eyring, V., Fischer, E., Fyfe, J., Friedlingstein, P., Knutti, R., Lowe, J., O'Neill, B., Sanderson, B., van Vuuren, D., Riahi, K., Meinshausen, M., Nicholls, Z., Tokarska, K.B., Hurtt, G., Kriegler, E., Lamarque, J.-F., Meehl, G., Moss, R., Bauer, S.E., Boucher, O., Brovkin, V., Byun, Y.-H., Dix, M., Gualdi, S., Guo, H., John, J.G., Kharin, S., Kim, Y., Koshiro, T., Ma, L., Olivié, D., Panickal, S., Qiao, F., Rong, X., Rosenbloom, N., Schupfner, M., Séférian, R., Sellar, A., Semmler, T., Shi, X., Song, Z., Steger, C., Stouffer, R., Swart, N., Tachiiri, K., Tang, Q., Tatebe, H., Voldoire, A., Volodin, E., Wyser, K., Xin, X., Yang, S., Yu, Y., Ziehn, T., 2021: Climate model projections from the Scenario Model Intercomparison Project (ScenarioMIP) of CMIP6. Earth System Dynamics 12, 253–293. https://doi.org/10.5194/esd-12-253-2021

**Response:** The use of the weather generator allowed us to scale the climate change signal of a climate simulation under a specific emission scenario to another emission scenario, based on a scaling factor (eq. 1, line 159) depending on the development of global mean temperatures (which depend on emission scenarios) simulated with the MAGICC model. We chose the RCP8.5 scenario (line 164-165) and will make this more clear in the text.

In addition, there is no serious critical assessment of the models selected for use in this work, particularly in relation to other potential sources of future climate model predictions mentioned above. For example, there is no serious evaluation of model bias associated with the different climate variables used to predict drought stress. How well do the selected models perform compared with more recent generations of climate model? Only generalised qualitative comments are made in this regard.

**Response:** We agree that a detailed model evaluation for the purpose of the study at hand should ideally be an integral part. However, with respect to our study it is difficult to define the important weather phenomena, which allow a selection of meteorologically reasonable climate models. For example, we made an evaluation of the impact of the length of dry spells on the occurrence of drought stress, but we found that this impact was in general small compared to the impact of the overall precipitation amount, since the soils serves as a storage for water. The complexity of the situation is enhanced by the diverse landscape characteristics and because the transition of moderate stress (at -0.3 MPa soil moisture tension respective predawn leaf water potential, positive for grapevines) to severe drought stress (at -0.6 MPa) corresponds to only 6-11 % change in available soil water capacity depending on the soil type. Since the authors from the climate-modelling field involved in this study contributed in kind (without funding), a detailed model evaluation proved to be infeasible.

We used a climate model ensemble already used in Maraun (2013) (this information is missing and we would add this to the revised paper). In this paper, the models were selected with the aim to separate the two sources of uncertainty, model errors and internal climate variability, regarding precipitation. In detail, three models from the MetOffice (UK) were excluded in our study, because the wind speed variable was on a different grid than the other weather variables (with possible implications on physical consistency of the weather variables). We concluded, in order to cover the uncertainty of future climate developments, this ensemble

would be adequate for our purpose. We suggest that we include this argumentation into the paper to clarify the path taken.

Douglas, M.: When will trends in European mean and heavy daily precipitation emerge? Environmental Research Letters, 8, 014004, http://dx.doi.org/10.1088/1748-9326/8/1/014004, 2013.

Concerning the critical assessment of the models, particularly in comparison with more recent climate simulations, we refer again to Kotlarski et al. (2014), where an evaluation of EURO-CORDEX and ENSEMBLES were performed for temperature and precipitation. In addition, the general comments in the discussion section (line 407-410) suggest that the projections of the used climate models are comparable with more recent climate simulations.

Line 407-410 of the discussion: "This bandwidth is comparable with the results of the REKLIES–DE project (±20 % for annual precipitation, region Germany and drainage basins of large rivers, 2070–2099 compared to 1971– 2000), calculated with 37 climate simulations including the EURO–CORDEX data (Hübener et al., 2017; Bülow et al., 2019). Additionally, similar seasonal shifts (increase of winter and decrease of summer precipitations) were reported in this study."

The evaluation of model bias associated with the different climate variables used to predict drought stress would need additional research. Model biases have different sources. In the case of our study, the climate variability within a grid box is also a potential source of a bias, because climate model data represent spatial means of the grid boxes. The grid box containing the weather station Geisenheim contains not only vineyards but also parts of the Taunus mountain range (about 200-400 m higher in altitude). In this part of the grid box, it is too cold to grow grapevines and annual precipitation is about 200 mm higher compared to the drier parts of the grid box, where viticulture is performed (we referred to this issue in the introduction, line 51-54, (see below) because we are very aware of this "problem"). Even if a climate simulation would perfectly simulate a spatial mean of a grid box of an observational climate, the grid box mean can have a substantial bias compared to the (point) data of a weather station contained in the grid box, an intrinsic problem of an approach to relate climate simulations on a larger scale to individual weather station data.

Line 51-54: 'Predictions on a high spatial resolution are a challenge in climate impact studies and mainly limited by the size of one grid box of regional climate models (RCMs). Although climatic conditions within a grid box may change from being suitable for vineyards to areas unsuitable for the cultivation of grapevines, climate change impact studies for European viticulture where often forced to be performed based on the spatial resolution of the underlying gridded climate model data.'

We observed substantial biases between climate model data and observed weather station data, when comparing historical periods (e.g. 1971-2000). We also observed spatial shifts of simulated precipitation patterns compared to observed precipitation patterns in the study region. At a small scale, the source of a bias of an individual climate model compared to data recorded by a weather station remains unclear. Using a bias as parameter to assess the quality/reliability of a projected climate change signal of a climate simulation would therefore need further research. Since we only used the change signals of the climate simulations, which are more stable at a small scale, climate model biases were less important for our study.

In general, impact models for grapevines like the used water budget model (or e.g. models for phenological development) were developed based on measurements taken in vineyards and weather stations located in or not far away of those vineyards. Therefore, for our case, the weather generator was the central tool to produce transient time series. There are two key features, on the one hand, the preservation of the statistics of observational weather patterns of a weather station at the transition from observed data to projected data, and on the other hand, the production of future time series incorporating the climate change signals of climate simulations.

We suggest to discuss our methodology of applying climate models with respect to our case study in more detail in comparison to other and thus different approaches to address the reviewers concerns.

2.3 Lack of clarity regarding spatial downscaling methods

The methodological steps from the 10 climate model predictions to the daily weather generator, and subsequently to the water balance model at vineyard scale could be more clearly described. A schematic flow diagram outlining the steps involved in the methodology in Section 2.2.2 would be helpful.

**Response:** We agree to the suggestion to add a schematic flow diagram to illustrate with more detail the downscaling steps from the climate data at 25 km resolution to the vineyard scale. We suggest describing the overall process more detailed in a supplement to the paper, including the schematic flow diagram. We also notice that the scaling of the climate change signals based on eq. 1, with the opportunity to scale the signals to different emission scenarios, is difficult to understand for the reader. An additional figure, showing the scaling factor as a function of the year, for different emission scenarios could help to better understand the methodology.

Section 2.2.2 seems to suggest that the regionally downscaled climate model data are provided at a spatial resolution of 25 km, and that these data then drive the weather generator at the same resolution. Is this resolution sufficient to provide realistic spatial variability within vineyard regions in complex terrain? I found that the progression from 25 km to vineyard scale climate predictions is not well explained. In Section 2.4 it is stated that 'The study was based on the high spatial resolution of individual plots.' (Line 201), and that the digital elevation model (DEM) data appear to be at 1 m spatial resolution, while soil information is at approximately 25 m resolution (see below).

Line: 397: 'The soil data go back mainly to soil mappings conducted from 1947–1958 (Böhm et al., 2007), where at distances of 20 m x 20 m, respectively 25 m x 25 m, soil samples down to 2 m depth were taken and analyses performed.'

**Response:** The weather generator was not driven at the same spatial resolution (see more details below in the next response) as the RCMs. The DEM (at 1 m resolution) was used to calculate the mean slope and aspect of a single vineyard. Slope and aspect are then included in the calculation of reference evapotranspiration (section 2.3, line 177-178). The soil information at approximately 25 m x 25 m was used to derive the necessary soil water storage capacity data to run the water balance model for each single vineyard (section 2.4, lines 205-213, but we admit that the spatial resolution information of the soil data is missing in this

section and will clarify this in the text). These two variables change substantially within the terrain and are important features of the water balance model approach.

Much of the subsequent analysis of results in the paper is based solely on the one weather station at Geisenheim, but there is also significant discussion of future drought stress in relation to individual vineyard plots (i.e. much finer resolution). It would be good to have a clearer explanation of how the model predictions of climate data at 25 km resolution are linked to the individual vineyard plots, presumably via assessment against weather station data and using the DEM and soil data in order to downscale to vineyard scale. For example, it is not entirely clear what is meant by the following statement:

Line 424: 'In order to downscale from the spatial means of grid box data of the RCMs to the spatial scale of station data, we used a weather generator to produce point data on the same scale as the weather stations and to simulate small–scale weather patterns'.

This statement suggests that climate variables from the regional climate models represent an average over 25 x 25 km grid squares (or volumes), but the underlined section above is unclear as 'weather station scale' is not defined. Figure 1 shows weather stations located within the two vineyard regions, often separated by only 2-5 km – is this what is meant by 'the same scale as the weather stations', or is 'weather station scale' a notional area represented by a single weather station (which may vary with terrain complexity)? If so, how is the weather generator used to downscale from 25 km resolution to 2-5 km resolution? Section 2.2.2 seems to be vague on this matter. In reality, the Rheingau vineyard region could be located within only one 25 x 25 km regional model grid cell. It is therefore unclear how the 12 or so weather stations located across the region are used to provide higher spatial resolution information in order to 'simulate small–scale weather patterns'.

**Response:** The climate simulations from 10 climate models were produced for each weather station (with observational data from 1959-1988) shown in Figure 1. A single vineyard plot (spatial resolution described in section 2.4, lines 200-205; and partially visible in Figures 11 and 12) was allocated to the nearest weather station. Therefore, each individual weather station represents an area defined by all vineyards that have the shortest distance to that weather station. We notice this information is missing in section 2.2.2 and we would add this to the revised manuscript. We think, with the vineyard plot specific data for reference evapotranspiration and soil data this approach is sufficient to provide the spatial variability of the terrain. We suggest an additional figure, maybe in a supplement to the manuscript, showing the time series of observed data (1961-1988) followed by the multi-model-mean for 1989-2100 for precipitation and/or the climatic water balance for all stations in order to illustrate the differences between the stations within the region.

Also, the statement on Line 441: '..we downscaled the grid box means of climate models to station (point) data in order to reduce the bias...' is vague and unhelpful, as it is obvious from the comments above that each grid box may contain several weather stations against which climate model output could be evaluated. It is therefore unclear how the model bias is assessed and/or reduced in this study.

**Response:** The Figures 6-9 show a smooth transition from the observed data of the weather station Geisenheim from 1961-1988 to the climate simulations from 1989-2100 and illustrate the reduction of the bias. Plotting the grid box data of the climate simulations directly would

result in sudden shifts (biases), different for each climate simulation, at the transition from 1988 to 1989 (the transition from observed to simulated data). An evaluation of the climate model output of the grid box means against the weather stations would only confirm the existence of a bias between those data. This is, from our point of view, not really necessary, if the methodical steps will be explained with more detail as suggested by the reviewer, which would then also illustrate with more detail, how the model bias is reduced.

The maps shown in Figures 11 and 12 suggest that a fine spatial resolution of drought stress was achieved, although the spatial resolution of the mapped data is not indicated in the caption.

**Response:** We will add this information of the spatial resolution in the caption.

As mentioned previously, maybe a schematic flow diagram would help to illustrate in detail the steps taken to downscale data from climate models to provide soil water information at vineyard scale.

**Response:** Thank you for this suggestion. We see from the comments above that relevant information is missing and a more detailed explanation of the downscaling methods is needed and will provide this information.

2.4 Lack of model validation

As mentioned previously, most of the results were presented for one site (Geisenheim), and no validation against other sites was shown. Although this study is '….applied to individual vineyard plots of two winegrowing regions….' (Line 508), there appears to be no real validation of the results at vineyard scale. A set of high-resolution maps is a produced (Figures 11 and 12), but the lack of validation against data from a range of weather station sites would be needed to assess their true value. Figures 6, 7, 8 & 9 indicate that there is significant overlap between the climate model data (1980s to 2100) and available observations for at least some regional climate stations (1980s to 2020), which should allow a comprehensive statistical analysis of model performance.

**Response:** The model validation needs to be separated into the validation of the weather generator, which produced the climate data (used for Figures 6-9) and the validation of the water balance model, which produced data about water balance of each single vineyard (Figures 11 and 12). The water balance model was developed with the aim to cover the most important variables affecting the water balance. It was validated for three single vineyards of the study region with different characteristics (slope and aspect, different usage of cover crops, row spacing and soil characteristics, see Hofmann et al., 2014, https://www.frontiersin.org/articles/10.3389/fpls.2014.00645/full). This approach should ensure that the water balance model could also be used for other vineyards in the region and that the calculated water balance developments were realistic. The model and previous versions thereof have been used and validated in different vineyard sites across Europe (i.e. Lebon et al., 2003; Pellegrino et al., 2006; Gaudin et al., 2014). A validation of the results at a larger vineyard scale (i.e. many different sites) is unfortunately not possible because the required water balance data are only available for a few vineyards (not area-wide) and observations.

References (not already mentioned in the paper):

Gaudin, R., Kansou, K., Payan, J.-C., Pellegrino, A., and Gary, C.: A water stress index based on water balance modelling for discrimination of grapevine quality and yield, OENO One, 48, 1-9, 10.20870/oeno-one.2014.48.1.1655, 2014.

Also, the validation results discussed in Section 3.1 are mostly subjective (e.g. '….no substantial bias of mean values or monthly sums between observed and synthetic values were apparent. (Lines 228-9)), and should be made more convincing through the use of rigorous statistical analysis to investigate more fully the differences between the distributions of observed and predicted variables (for a number of climate stations). Otherwise, it is not possible for the reader to properly assess the efficacy of the model downscaling and evaluate the conclusions reached in this study.

**Response:** The validation results in section 3.1 refer to the weather generator (WG) and its capability to reproduce a climate observed from weather stations. To calibrate the WG, observed weather data from 1959-1988 (the 'baseline climate') were used. We agree to improve the statistical analysis and to include more weather stations.

Related to the previous comment, it would have been useful to comment more fully on the results shown in Figure 2. The synthetic data in this figure show lower rainfall, higher evapotranspiration and higher solar radiation compared with observations, in addition to the smaller range of their frequency distributions. Assuming that the model predictions are correct, is it possible that this reflects a general change in weather patterns under the selected scenario from cloudy low-pressure systems to clearer high-pressure systems? If so, what other climate risks could be associated with such a trend (e.g. increased frost frequency)?

**Response:** The synthetic data of Figure 2 do not include model predictions. The weather generator parameters describing the statistical structure of the observed climate (derived from the observed climate, 1959-1988) were not modified. We will try to improve the text for clarity.

2.5 Scenarios unclear

Although both the A1B and RCP8.5 scenarios are mentioned in Section 2.2.2, there is no indication of which scenario is used in the subsequent analysis sections (until Section 4 – Discussion). A significant omission is that none of the figures in the results sections mention the scenario that has been applied to achieve the results shown in each figure (it should be included in the captions). It should also have been emphasised that the RCP8.5 scenario represents 'business as usual' and is therefore the most extreme emissions scenario. Comparative maps of different scenarios (e.g. RCP4.5 and RCP 8.5) would be an interesting addition, alongside evaluation of any differences in the seasonality of drought risk that might occur under different scenarios. Referring to other studies, it is mentioned that 'Noteworthy, the projected bandwidth for precipitation for the mitigation scenario RCP2.6 are less than half of those for RCP8.5 (Hübener et al., 2017).' (Lines 416-417), but there is no attempt to undertake such a comparison between scenarios in this study.

**Response:** It is mentioned at the end of section 2.2.2 (line 164-165) that we chose the high baseline emission scenario RCP8.5. However, we agree that a comparative map for a different

scenario could improve the results and that the emission scenario RCP8.5 needs to be discussed in more detail. Recent literature (Hausfather and Peters, 2020; Burgess et al., 2021) also suggests that RCP8.5 is increasingly implausible because it requires a very high and increasing coal use, which diverges from observed trends and energy projections of global $CO_2$ emissions.

Hausfather, Z., and Peters, G. P.: Emissions - the 'business as usual' story is misleading, Nature, 577, 618-620, https://doi.org/10.1038/d41586-020-00177-3, 2020.

Burgess, M. G., Ritchie, J., Shapland, J., and Pielke, R.: IPCC baseline scenarios have over-projected CO2 emissions and economic growth, Environmental Research Letters, 16, 014016, https://doi.org/10.1088/1748-9326/abcdd2, 2020.

2.6 Statistical interpretation

There is no detailed interpretation of the p-value trends shown in Figures 6b and 7b, only the brief statements:

Lines 290-1: 'For seven simulations, the projected trends were significant after the year 2073 (Mann–Kendall trend test, $p < 0.05$, Fig. 6b).'

Lines 292-3: 'The statistical significance of the trends was comparable to the trends of precipitation (Fig. 7b).'

Presumably, the null hypothesis being tested is that predicted precipitation trends are no different from zero, but the trends in p-values for the 10 models for both annual precipitation and climate water balance are only very briefly discussed. It seems to me that until about 2030 most models show no trend in precipitation, while by about 2070 eight out of ten models appear to show a statistically significant trend (in a couple of cases a negative trend). A major shift seems to take place between about 2030 and 2050. In contrast, the results for the climate water balance shown in Figure 7 seem markedly different, with only two models showing a statistically significant trend by about 2030, and much less agreement between models as to future trends. It would be useful to have further discussion of likely mechanisms here (in Section 3.3.1).

**Response:** Thank you for pointing this out. The contrast between the precipitation trends (Fig. 6) and trends of the climate water balance (Fig. 7) is related to the increase in reference evapotranspiration (see Table 3, line 337). We agree to discuss this more detailed.

Similarly, it would be useful to have more critical analysis of the results shown in Figure 10. The remarkable difference between the potential drought stress for the two periods (1989-2018 and 2041-2070) is not adequately explained. Presumably, the wide range of values shown for 2041-2070 for both regions could be explained by three poor-performing models, and if they were removed the differences between 1989-2018 and 2041-2070 may actually be minimal (but there is no such critical analysis here). There are some rather vague qualitative comparisons of 'bandwidth' in modelled precipitation mentioned in Section 4 (Discussion), and in relation to model evaluation, it is stated that 'This bandwidth could be reduced if the extreme models at the upper or lower edge would be excluded, but since no direct model flaws were detected, this would exclude possible future climate realisations.' (Line 411-12).

However, based on the information provided in the paper there does not seem to have been any serious attempt to undertake model validation (and I am not sure what a 'direct model flaw' is). There therefore seems to have been a lack of detailed critical analysis of the rather dated climate models used in this study, as mentioned earlier, and this seems to be a major weakness of this work.

**Response:** We focused on describing the uncertainty of the ensemble in terms of possible future climate developments rather than discussing individual models of the ensemble. A serious validation of the climate models, which would maybe end up in an exclusion of a climate simulation, is critical and this could not be performed for the reasons of feasibility mentioned above. What we meant with "direct model flaw" could also refer to errata of climate data, usage restrictions or other reported issues. Those issues could lead to an exclusion or withdrawal of a model in a project, for instance as reported by the Reklies-De project (see point 4. of http://reklies.wdc-climate.de/ but only in German). We suggest formulating "severe shortcomings" instead of "direct model flaw". To the knowledge of the authors such shortcomings regarding the used climate models were not reported elsewhere.

2.7 Lack of future research directions

There is no clear statement in the discussion outlining where this research might lead and what topics would be worth following up.

**Response:** We will add more information concerning this point, for both, the viticultural as well as the climate modelling perspective.

3. **Technical corrections, including typing errors and English expression**

There are a lot of problems with basic English expression which in some cases make the explanations confusing. Some suggested changes are indicated below:

**Response:** We appreciate all corrections and will incorporate them into the revised manuscript.

In several places:

Replace 'row distance' by 'row spacing'

Replace 'approx.' by 'approximately'

I would also suggest that the word 'bandwidth' is replaced with 'uncertainty' or 'variability' throughout the text as it provides a better indication of its significance in this application.

Line 11: Fix punctuation to – 'Extended periods without precipitation, observed for example in Central Europe including Germany during the seasons from 2018 to 2020, can lead to water deficit…..'

Lines 22-23: Replace 'Possible adaptation measures depend highly on local conditions and to make targeted use of the resource water,….' with 'Possible adaptation measures depend highly on local conditions and are needed to make targeted use of the resource water, while….'

Line 34: Replace 'Soil moisture decreased across Europe…..' with 'Soil moisture has decreased across Europe…..'

Line 38: Replace 'Despite of some newly emerging wine regions….' with 'Despite some newly emerging wine regions….'

Line 39: Replace '…..economically important grape cultivation of Europe.' with 'economically important grape cultivation in Europe.'

Line 50: Replace '…..cover crop or canopy management up to the implementation of irrigation systems.' with '…..cover crop or canopy management as well as the implementation of irrigation systems.'

Line 51: Replace 'Predictions on a high spatial resolution…' with 'High spatial resolution predictions…'

Line 53: Replace 'where' with 'were'

Line 57: Replace '….. the latter one also included possible changes in interannual variability.' with '….. with the latter study also including possible changes in interannual variability.'

Lines 61-64: Replace '….Moriondo et al. (2010) for expected changes for the premium wine quality area of Tuscany on a fine spatial resolution (1 km x 1 km, based on downscaling climate projections to station data and spatial interpolation). Only a few studies used data from soil maps including AWC as input data (Fraga et al., 2013; Moriondo et al., 2013), but on a spatial resolution still to coarse to represent the heterogeneity within growing regions. Recently, fine scale variability within growing regions were assessed…..' with '….Moriondo et al. (2010) for expected changes in the premium wine quality area of Tuscany at a fine spatial resolution (1 km x 1 km, based on downscaling climate projections to station scale using spatial interpolation). Only a few studies used data from soil maps that included AWC as input data (Fraga et al., 2013; Moriondo et al., 2013), but often at a spatial resolution still too coarse to represent the heterogeneity within growing regions. Recently, fine scale variability within growing regions has been assessed…..'

Lines 69-70: Replace 'Especially AWC, slope and aspect are very heterogeneous in steep slope regions and thus is the supply of and demand for water.' with 'AWC, slope and aspect are particularly heterogeneous in regions of complex terrain resulting in variability in the supply of and demand for water.'

Line 72: Replace 'like' with 'such as'

Line 73: Replace 'adapted' with 'modified'

Line 78: Replace '….the main objective of the study was to quantify the likelihood of risk for future water deficit on…' with '….the main objective of this study is to quantify the likelihood of risk of future water deficit on….'

Line 81: Replace '… for the characterization of vineyard landscapes…' with '… in order to characterize vineyard landscapes…'

Line 90: Replace '… bounded by the Rhine river to the south and the ridge of the Taunus mountain range in the north, and the vineyards near….' with '… bounded by the Rhine river to the south and the ridge of the Taunus mountain range in the north, as well as the vineyards near….'

Line 95: Replace '…in the west…' with '…. to the west…'

Line 99: Replace 'at' with 'on'

Line 102: Replace 'predominant' with 'particularly'

Line 109: Delete 'as'

Line 110: Delete 'was'

Line 118: Replace 'We worked with four time series, two observed and two synthetic series.' With 'We worked with four time series, two observed and two synthetic.'

Line 121-123: The sense of the sentence 'All stations recorded precipitation and the station Bensheim additionally temperature and relative humidity (also used for the other stations at the Hessische Bergstraße).' is unclear.

Table 1: Some column headings need reformatting.

Lines 184 & 185: Replace '…in form of…' with '…. in the form of….'

Lines 197-199: Poor English expression.

Line 215: Replace '… to assess…' with '…of …'

Line 216: Replace '…as drought…' with '…as a drought …'

Line 229: Replace '…on the other side…' with '…on the other hand …'

Line 233: Replace '…. for dry years…' with '…. of dry years…'

[There are many more, but I don't have time to correct them all. I suggest that a native English editor is used to eliminate the remaining issues.]

 Referencing errors:

Line 141: Replace 'Garofalo et al., 2018' by 'Garofalo et al., 2019'

Lines 483 and 492:  Replace 'Van Leeuwen et al., 2017' by 'Van Leeuwen, C., and Destrac-Irvine, 2017'

---

## Author Response (AR1)

Dear Editor and Referees,

We would like to thank the anonymous referees for their time and constructive comments, which helped to improve the manuscript. We have addressed all the comments and revised the manuscript accordingly. Briefly summarised, our main revisions concern the inclusion of the RCP4.5 emission scenario and further details on what is driving the changes in reference evapotranspiration, which are both presented in the supplement to the paper. The sections 3.1, validation of the weather generator, and 3.3.1, projected annual trends of precipitation and the climatic water balance to 2100, were intensively revised. In the following, we take up again the comments of the referees and our answers, as in the documents https://doi.org/10.5194/esd-2021-9-AC1 and https://doi.org/10.5194/esd-2021-9-AC2, and describe here point by point, where we have made changes in the manuscript (in green font). The line numbers in our comments refer to the marked-up manuscript version.

**Referee #1:**

1. **General comments**

The rationale of the paper is based on the need to improve knowledge of local climate and soil moisture in order for vineyard regions to respond appropriately to climate change. The paper emphasises the need to downscale from global climate model predictions using regional climate models in order to assess effects on the water budget of grapevines at vineyard scale, so is in line with contemporary research trends. A weather generator and a water balance model are also used, accounting for local variations in soil characteristics, the complexity of the terrain, and crop management practices, in order to provide predictions of drought risk at vineyard scale.

The overall aim of the paper is therefore in line with objectives of international research into the application of climate models to assess impacts of and develop appropriate responses to climate change by downscaling model projections of future climate to vineyard scale. Although the general approach is fine, there are several areas of weakness in the paper, as discussed in the following section. In particular, some aspects of the methodology used in the paper seem to be rather dated and not clearly described. In particular, I would have expected that more recent climate models would have been used, given the rapid climate model development that has taken place over the past decade.

2. **Specific comments**

2.1 Soil moisture versus temperature

Line 31-32: I disagree with the general statement that 'Within the existing production areas, water shortage is probably the most dominant environmental constraint (Williams and Matthews, 1990) ....', which the authors appear to suggest applies globally. In many parts of the world, it is clear that temperature has a greater impact on grape production and wine quality, especially in 'New World' regions where irrigation is a standard practice.

**Response:** We agree that temperature has a greater impact on grape production and wine quality than water availability. Nevertheless, water is an important limiting factor not only

under irrigated conditions. The statement "within the existing production areas" refers to areas, where temperature conditions are not a limiting factor for the cultivation of grapevines. The need to irrigate in 'New World' regions is an example that water shortage is an environmentally limiting factor within those regions. To avoid misunderstandings, we suggest to write: 'Within the existing production areas, where temperature conditions are in general favourable for cultivation, water shortage is probably the most dominant environmental constraint (Williams and Matthews, 1990)… '

**Changes in the manuscript:** We modified the sentence as suggested in our comment (line 32).

2.2 Dated climate models

The climate models used in this research appear to be quite old and outdated (van der Linden and Mitchell, 2009) given the rapid developments in model design and downscaling techniques over the past decade. Even the web link for the ENSEMBLES Final Report states 'This object has been archived because its content is outdated.' (https://climate-adapt.eea.europa.eu/metadata/publications/ensembles-final-report). It is therefore unclear why more recent climate models from the CMIP5 and CMIP6 evaluations, or the available EURO-CORDEX model data are not used in this work. Recent publications referenced in this paper (e.g. Gutiérrez et al. 2019) appear to suggest that EURO-CORDEX is the preferred model framework for contemporary research, and there are many publications over the past decade that have been based on CORDEX climate model data.

**Response:** We agree that using a regional climate model ensemble from the ENSEMBLES project raises the question why data from EURO-CORDEX (the successor of ENSEMBLES) were not used in this study. When we started with the project (in 2015), all nodes of the Earth System Grid Federation (ESGF) to download the EURO-CORDEX data were out of service for several months without information of a date of return. The evaluation of the performance of the climate models against observational data of Kotlarski et al. (2014) showed that the improvements of EURO-CORDEX compared to ENSEMBLES were not very significant (a detailed comparison is described in section 4.6 of this paper). Kotlarski et al. reported comparable bias ranges for EURO-CORDEX and the ENSEMBLES simulations. Since we used a weather generator in our study, we needed only the change signals of the climate simulations. High-resolution RCM ensemble simulations of Feldmann et al. (2013) showed that the relative change of mean precipitation is quite uniform in the study region (with the limitation that Feldmann et al. focused on the near future 2011-2040), but suggesting that also the higher spatial resolution of EURO-CORDEX (12 km vs. 25 km) is of limited added value for our study. Overall, we concluded that the use the ENSEMBLES instead of EURO-CORDEX would not have a significant effect on the results, and, because we also had to start with the work, we decided to use the ENSEMBLES simulations. We suggest to include parts of the argumentation above in the paper to make it clear to the reader why the used models have their value.

Feldmann, H., Schädler, G., Panitz, H.-J., and Kottmeier, C.: Near future changes of extreme precipitation over complex terrain in Central Europe derived from high resolution RCM ensemble simulations, International Journal of Climatology, 33, 1964-1977, 10.1002/joc.3564, 2013.

Kotlarski, S., Keuler, K., Christensen, O. B., Colette, A., Déqué, M., Gobiet, A., Goergen, K., Jacob, D., Lüthi, D., van Meijgaard, E., Nikulin, G., Schär, C., Teichmann, C., Vautard, R.,

Warrach-Sagi, K., and Wulfmeyer, V.: Regional climate modeling on European scales: a joint standard evaluation of the EURO-CORDEX RCM ensemble, Geosci. Model Dev., 7, 1297-1333, 10.5194/gmd-7-1297-2014, 2014.

**Changes in the manuscript:** We included the main points of the argumentation at the end of section 2.2.2 (lines 180-183).

The dated nature of the climate modelling component of this work is also evident by the reference in Section 2.2.2 to application of the 10 selected models to the old A1B emission scenario (Line 152), which was developed over 20 years ago and has since been replaced by RCP scenarios (about ten years ago) and more recently by SSP scenarios (Tebaldi et al. 2021 – see below).

Tebaldi, C., Debeire, K., Eyring, V., Fischer, E., Fyfe, J., Friedlingstein, P., Knutti, R., Lowe, J., O'Neill, B., Sanderson, B., van Vuuren, D., Riahi, K., Meinshausen, M., Nicholls, Z., Tokarska, K.B., Hurtt, G., Kriegler, E., Lamarque, J.-F., Meehl, G., Moss, R., Bauer, S.E., Boucher, O., Brovkin, V., Byun, Y.-H., Dix, M., Gualdi, S., Guo, H., John, J.G., Kharin, S., Kim, Y., Koshiro, T., Ma, L., Olivié, D., Panickal, S., Qiao, F., Rong, X., Rosenbloom, N., Schupfner, M., Séférian, R., Sellar, A., Semmler, T., Shi, X., Song, Z., Steger, C., Stouffer, R., Swart, N., Tachiiri, K., Tang, Q., Tatebe, H., Voldoire, A., Volodin, E., Wyser, K., Xin, X., Yang, S., Yu, Y., Ziehn, T., 2021: Climate model projections from the Scenario Model Intercomparison Project (ScenarioMIP) of CMIP6. Earth System Dynamics 12, 253–293. https://doi.org/10.5194/esd-12-253-2021

**Response:** The use of the weather generator allowed us to scale the climate change signal of a climate simulation under a specific emission scenario to another emission scenario, based on a scaling factor (eq. 1, line 159) depending on the development of global mean temperatures (which depend on emission scenarios) simulated with the MAGICC model. We chose the RCP8.5 scenario (line 164-165) and will make this more clear in the text.

**Changes in the manuscript:** We have included additional simulations for RCP4.5 and made the use of RCP8.5 and RCP4.5 more clear in the text (lines 174-176). To better describe the methodology we have added a graph in the supplement (Fig. S2), showing the temporal evolution of the scaling factor $k$ for different emission scenarios.

In addition, there is no serious critical assessment of the models selected for use in this work, particularly in relation to other potential sources of future climate model predictions mentioned above. For example, there is no serious evaluation of model bias associated with the different climate variables used to predict drought stress. How well do the selected models perform compared with more recent generations of climate model? Only generalised qualitative comments are made in this regard.

**Response:** We agree that a detailed model evaluation for the purpose of the study at hand should ideally be an integral part. However, with respect to our study it is difficult to define the important weather phenomena, which allow a selection of meteorologically reasonable climate models. For example, we made an evaluation of the impact of the length of dry spells on the occurrence of drought stress, but we found that this impact was in general small compared to the impact of the overall precipitation amount, since the soils serves as a storage for water. The complexity of the situation is enhanced by the diverse landscape characteristics and because the transition of moderate stress (at -0.3 MPa soil moisture tension respective

predawn leaf water potential, positive for grapevines) to severe drought stress (at -0.6 MPa) corresponds to only 6-11 % change in available soil water capacity depending on the soil type. Since the authors from the climate-modelling field involved in this study contributed in kind (without funding), a detailed model evaluation proved to be infeasible.

We used a climate model ensemble already used in Maraun (2013) (this information is missing and we would add this to the revised paper). In this paper, the models were selected with the aim to separate the two sources of uncertainty, model errors and internal climate variability, regarding precipitation. In detail, three models from the MetOffice (UK) were excluded in our study, because the wind speed variable was on a different grid than the other weather variables (with possible implications on physical consistency of the weather variables). We concluded, in order to cover the uncertainty of future climate developments, this ensemble would be adequate for our purpose. We suggest that we include this argumentation into the paper to clarify the path taken.

Douglas, M.: When will trends in European mean and heavy daily precipitation emerge? Environmental Research Letters, 8, 014004, http://dx.doi.org/10.1088/1748-9326/8/1/014004, 2013.

Concerning the critical assessment of the models, particularly in comparison with more recent climate simulations, we refer again to Kotlarski et al. (2014), where an evaluation of EURO-CORDEX and ENSEMBLES were performed for temperature and precipitation. In addition, the general comments in the discussion section (line 407-410) suggest that the projections of the used climate models are comparable with more recent climate simulations.

Line 407-410 of the discussion: "This bandwidth is comparable with the results of the REKLIES–DE project (±20 % for annual precipitation, region Germany and drainage basins of large rivers, 2070–2099 compared to 1971– 2000), calculated with 37 climate simulations including the EURO–CORDEX data (Hübener et al., 2017; Bülow et al., 2019). Additionally, similar seasonal shifts (increase of winter and decrease of summer precipitations) were reported in this study."

The evaluation of model bias associated with the different climate variables used to predict drought stress would need additional research. Model biases have different sources. In the case of our study, the climate variability within a grid box is also a potential source of a bias, because climate model data represent spatial means of the grid boxes. The grid box containing the weather station Geisenheim contains not only vineyards but also parts of the Taunus mountain range (about 200-400 m higher in altitude). In this part of the grid box, it is too cold to grow grapevines and annual precipitation is about 200 mm higher compared to the drier parts of the grid box, where viticulture is performed (we referred to this issue in the introduction, line 51-54, (see below) because we are very aware of this "problem"). Even if a climate simulation would perfectly simulate a spatial mean of a grid box of an observational climate, the grid box mean can have a substantial bias compared to the (point) data of a weather station contained in the grid box, an intrinsic problem of an approach to relate climate simulations on a larger scale to individual weather station data.

Line 51-54: 'Predictions on a high spatial resolution are a challenge in climate impact studies and mainly limited by the size of one grid box of regional climate models (RCMs). Although climatic conditions within a grid box may change from being suitable for vineyards to areas unsuitable for the cultivation of grapevines, climate change impact studies for European

viticulture where often forced to be performed based on the spatial resolution of the underlying gridded climate model data.'

We observed substantial biases between climate model data and observed weather station data, when comparing historical periods (e.g. 1971-2000). We also observed spatial shifts of simulated precipitation patterns compared to observed precipitation patterns in the study region. At a small scale, the source of a bias of an individual climate model compared to data recorded by a weather station remains unclear. Using a bias as parameter to assess the quality/reliability of a projected climate change signal of a climate simulation would therefore need further research. Since we only used the change signals of the climate simulations, which are more stable at a small scale, climate model biases were less important for our study.

In general, impact models for grapevines like the used water budget model (or e.g. models for phenological development) were developed based on measurements taken in vineyards and weather stations located in or not far away of those vineyards. Therefore, for our case, the weather generator was the central tool to produce transient time series. There are two key features, on the one hand, the preservation of the statistics of observational weather patterns of a weather station at the transition from observed data to projected data, and on the other hand, the production of future time series incorporating the climate change signals of climate simulations.

We suggest to discuss our methodology of applying climate models with respect to our case study in more detail in comparison to other and thus different approaches to address the reviewers concerns.

**Changes in the manuscript:** We have added references to studies, where the used climate models were evaluated (lines 184-186).

2.3 Lack of clarity regarding spatial downscaling methods

The methodological steps from the 10 climate model predictions to the daily weather generator, and subsequently to the water balance model at vineyard scale could be more clearly described. A schematic flow diagram outlining the steps involved in the methodology in Section 2.2.2 would be helpful.

**Response:** We agree to the suggestion to add a schematic flow diagram to illustrate with more detail the downscaling steps from the climate data at 25 km resolution to the vineyard scale. We suggest describing the overall process more detailed in a supplement to the paper, including the schematic flow diagram. We also notice that the scaling of the climate change signals based on eq. 1, with the opportunity to scale the signals to different emission scenarios, is difficult to understand for the reader. An additional figure, showing the scaling factor as a function of the year, for different emission scenarios could help to better understand the methodology.

**Changes in the manuscript:** We have added two schematic flow diagrams to the supplement of the manuscript (Fig. S1 and Fig. S3) as suggested by the referee. The first describes the steps to generate the station-specific climate simulations and the second one the steps to the water balance simulation at vineyard scale.

Section 2.2.2 seems to suggest that the regionally downscaled climate model data are provided at a spatial resolution of 25 km, and that these data then drive the weather generator at the same resolution. Is this resolution sufficient to provide realistic spatial variability within vineyard regions in complex terrain? I found that the progression from 25 km to vineyard scale climate predictions is not well explained. In Section 2.4 it is stated that 'The study was based on the high spatial resolution of individual plots.' (Line 201), and that the digital elevation model (DEM) data appear to be at 1 m spatial resolution, while soil information is at approximately 25 m resolution (see below).

Line: 397: 'The soil data go back mainly to soil mappings conducted from 1947–1958 (Böhm et al., 2007), where at distances of 20 m x 20 m, respectively 25 m x 25 m, soil samples down to 2 m depth were taken and analyses performed.'

**Response:** The weather generator was not driven at the same spatial resolution (see more details below in the next response) as the RCMs. The DEM (at 1 m resolution) was used to calculate the mean slope and aspect of a single vineyard. Slope and aspect are then included in the calculation of reference evapotranspiration (section 2.3, line 177-178). The soil information at approximately 25 m x 25 m was used to derive the necessary soil water storage capacity data to run the water balance model for each single vineyard (section 2.4, lines 205-213, but we admit that the spatial resolution information of the soil data is missing in this section and will clarify this in the text). These two variables change substantially within the terrain and are important features of the water balance model approach.

**Changes in the manuscript:** We have modified the section 2.4 and moved the information of the spatial resolution of the soil data from the discussion section (line 535) to section 2.4 (lines 235-236). Referee#2 asked for clarifications regarding the rooting depth of the region, which is also mentioned here (lines 236-240). The calculation of reference evapotranspiration taking into account the slope and aspect of the vineyards is additionally mentioned in the flow diagram (Fig. S3).

Much of the subsequent analysis of results in the paper is based solely on the one weather station at Geisenheim, but there is also significant discussion of future drought stress in relation to individual vineyard plots (i.e. much finer resolution). It would be good to have a clearer explanation of how the model predictions of climate data at 25 km resolution are linked to the individual vineyard plots, presumably via assessment against weather station data and using the DEM and soil data in order to downscale to vineyard scale. For example, it is not entirely clear what is meant by the following statement:

Line 424: 'In order to downscale from the spatial means of grid box data of the RCMs to the spatial scale of station data, we used a weather generator to produce point data on the same scale as the weather stations and to simulate small–scale weather patterns'.

This statement suggests that climate variables from the regional climate models represent an average over 25 x 25 km grid squares (or volumes), but the underlined section above is unclear as 'weather station scale' is not defined. Figure 1 shows weather stations located within the two vineyard regions, often separated by only 2-5 km – is this what is meant by 'the same scale as the weather stations', or is 'weather station scale' a notional area represented by a single weather station (which may vary with terrain complexity)? If so, how is the weather generator used to downscale from 25 km resolution to 2-5 km resolution? Section 2.2.2 seems

to be vague on this matter. In reality, the Rheingau vineyard region could be located within only one 25 x 25 km regional model grid cell. It is therefore unclear how the 12 or so weather stations located across the region are used to provide higher spatial resolution information in order to 'simulate small–scale weather patterns'.

**Response:** The climate simulations from 10 climate models were produced for each weather station (with observational data from 1959-1988) shown in Figure 1. A single vineyard plot (spatial resolution described in section 2.4, lines 200-205; and partially visible in Figures 11 and 12) was allocated to the nearest weather station. Therefore, each individual weather station represents an area defined by all vineyards that have the shortest distance to that weather station. We notice this information is missing in section 2.2.2 and we would add this to the revised manuscript. We think, with the vineyard plot specific data for reference evapotranspiration and soil data this approach is sufficient to provide the spatial variability of the terrain. We suggest an additional figure, maybe in a supplement to the manuscript, showing the time series of observed data (1961-1988) followed by the multi-model-mean for 1989-2100 for precipitation and/or the climatic water balance for all stations in order to illustrate the differences between the stations within the region.

**Changes in the manuscript:** We have added the information about the allocation of the vineyard plots to the weather stations in section 2.4 (lines 243-245) and it is also mentioned in the flow diagram (Fig. S3 in the Supplement). We have also added the suggested figure about observed and projected precipitation data to the supplement (Fig. S5) and referred to the figure in line 244. We have deleted the sentence 'In order to downscale from the spatial means of grid box data of the RCMs to the spatial scale of station data, we used a weather generator to produce point data on the same scale as the weather stations and to simulate small–scale weather patterns' (lines 562-564) as it is confusing and no longer needed.

Also, the statement on Line 441: '..we downscaled the grid box means of climate models to station (point) data in order to reduce the bias...' is vague and unhelpful, as it is obvious from the comments above that each grid box may contain several weather stations against which climate model output could be evaluated. It is therefore unclear how the model bias is assessed and/or reduced in this study.

**Response:** The Figures 6-9 show a smooth transition from the observed data of the weather station Geisenheim from 1961-1988 to the climate simulations from 1989-2100 and illustrate the reduction of the bias. Plotting the grid box data of the climate simulations directly would result in sudden shifts (biases), different for each climate simulation, at the transition from 1988 to 1989 (the transition from observed to simulated data). An evaluation of the climate model output of the grid box means against the weather stations would only confirm the existence of a bias between those data. This is, from our point of view, not really necessary, if the methodical steps will be explained with more detail as suggested by the reviewer, which would then also illustrate with more detail, how the model bias is reduced.

**Changes in the manuscript:** We think, that with the supplementary information about the methodology, the reader can now better understand, how the change signals of the climate models are represented in the station-specific climate simulations. We agree that the statement '..we downscaled the grid box means of climate models to station (point) data in

order to reduce the bias…' is unhelpful and have replaced it with a reference to the weather generator approach (lines 592-594).

The maps shown in Figures 11 and 12 suggest that a fine spatial resolution of drought stress was achieved, although the spatial resolution of the mapped data is not indicated in the caption.

**Response:** We will add this information of the spatial resolution in the caption.

**Changes in the manuscript:** The information is added (lines 514 and 521).

As mentioned previously, maybe a schematic flow diagram would help to illustrate in detail the steps taken to downscale data from climate models to provide soil water information at vineyard scale.

**Response:** Thank you for this suggestion. We see from the comments above that relevant information is missing and a more detailed explanation of the downscaling methods is needed and will provide this information.

**Changes in the manuscript:** We refer to the comments above and agree that the schematic flow diagrams have improved the manuscript.

2.4 Lack of model validation

As mentioned previously, most of the results were presented for one site (Geisenheim), and no validation against other sites was shown. Although this study is '….applied to individual vineyard plots of two winegrowing regions….' (Line 508), there appears to be no real validation of the results at vineyard scale. A set of high-resolution maps is a produced (Figures 11 and 12), but the lack of validation against data from a range of weather station sites would be needed to assess their true value. Figures 6, 7, 8 & 9 indicate that there is significant overlap between the climate model data (1980s to 2100) and available observations for at least some regional climate stations (1980s to 2020), which should allow a comprehensive statistical analysis of model performance.

**Response:** The model validation needs to be separated into the validation of the weather generator, which produced the climate data (used for Figures 6-9) and the validation of the water balance model, which produced data about water balance of each single vineyard (Figures 11 and 12). The water balance model was developed with the aim to cover the most important variables affecting the water balance. It was validated for three single vineyards of the study region with different characteristics (slope and aspect, different usage of cover crops, row spacing and soil characteristics, see Hofmann et al., 2014, https://www.frontiersin.org/articles/10.3389/fpls.2014.00645/full). This approach should ensure that the water balance model could also be used for other vineyards in the region and that the calculated water balance developments were realistic. The model and previous versions thereof have been used and validated in different vineyard sites across Europe (i.e. Lebon et al., 2003; Pellegrino et al., 2006; Gaudin et al., 2014). A validation of the results at a larger vineyard scale (i.e. many different sites) is unfortunately not possible because the required water balance data are only available for a few vineyards (not area-wide) and observations.

References (not already mentioned in the paper):

Gaudin, R., Kansou, K., Payan, J.-C., Pellegrino, A., and Gary, C.: A water stress index based on water balance modelling for discrimination of grapevine quality and yield, OENO One, 48, 1-9, 10.20870/oeno-one.2014.48.1.1655, 2014.

**Changes in the manuscript:** We have rewritten section 3.1 completely (lines 258-294). In addition to the comparisons of seasonal and annual distributions in Fig. 2, we have added distributions of daily data, power spectra (suggested by referee#2) and an indirect validation by comparing water balance calculations of observed and synthetic data of the weather generator for three different weather stations. We think, that we have now presented the strength and weaknesses of the approach in a more comprehensible way.

Also, the validation results discussed in Section 3.1 are mostly subjective (e.g. '....no substantial bias of mean values or monthly sums between observed and synthetic values were apparent. (Lines 228-9)), and should be made more convincing through the use of rigorous statistical analysis to investigate more fully the differences between the distributions of observed and predicted variables (for a number of climate stations). Otherwise, it is not possible for the reader to properly assess the efficacy of the model downscaling and evaluate the conclusions reached in this study.

**Response:** The validation results in section 3.1 refer to the weather generator (WG) and its capability to reproduce a climate observed from weather stations. To calibrate the WG, observed weather data from 1959-1988 (the 'baseline climate') were used. We agree to improve the statistical analysis and to include more weather stations.

**Changes in the manuscript:** Please see the revised section 3.1 and the comment above.

Related to the previous comment, it would have been useful to comment more fully on the results shown in Figure 2. The synthetic data in this figure show lower rainfall, higher evapotranspiration and higher solar radiation compared with observations, in addition to the smaller range of their frequency distributions. Assuming that the model predictions are correct, is it possible that this reflects a general change in weather patterns under the selected scenario from cloudy low-pressure systems to clearer high-pressure systems? If so, what other climate risks could be associated with such a trend (e.g. increased frost frequency)?

**Response:** The synthetic data of Figure 2 do not include model predictions. The weather generator parameters describing the statistical structure of the observed climate (derived from the observed climate, 1959-1988) were not modified. We will try to improve the text for clarity.

2.5 Scenarios unclear

Although both the A1B and RCP8.5 scenarios are mentioned in Section 2.2.2, there is no indication of which scenario is used in the subsequent analysis sections (until Section 4 – Discussion). A significant omission is that none of the figures in the results sections mention the scenario that has been applied to achieve the results shown in each figure (it should be included in the captions). It should also have been emphasised that the RCP8.5 scenario represents 'business as usual' and is therefore the most extreme emissions scenario.

Comparative maps of different scenarios (e.g. RCP4.5 and RCP 8.5) would be an interesting addition, alongside evaluation of any differences in the seasonality of drought risk that might occur under different scenarios. Referring to other studies, it is mentioned that 'Noteworthy, the projected bandwidth for precipitation for the mitigation scenario RCP2.6 are less than half of those for RCP8.5 (Hübener et al., 2017).' (Lines 416-417), but there is no attempt to undertake such a comparison between scenarios in this study.

**Response:** It is mentioned at the end of section 2.2.2 (line 164-165) that we chose the high baseline emission scenario RCP8.5. However, we agree that a comparative map for a different scenario could improve the results and that the emission scenario RCP8.5 needs to be discussed in more detail. Recent literature (Hausfather and Peters, 2020; Burgess et al., 2021) also suggests that RCP8.5 is increasingly implausible because it requires a very high and increasing coal use, which diverges from observed trends and energy projections of global $CO_2$ emissions.

Hausfather, Z., and Peters, G. P.: Emissions - the 'business as usual' story is misleading, Nature, 577, 618-620, https://doi.org/10.1038/d41586-020-00177-3, 2020.

Burgess, M. G., Ritchie, J., Shapland, J., and Pielke, R.: IPCC baseline scenarios have over-projected CO2 emissions and economic growth, Environmental Research Letters, 16, 014016, https://doi.org/10.1088/1748-9326/abcdd2, 2020.

**Changes in the manuscript:** We have mentioned the emission scenarios more clearly in section 2.2.2 (lines 174-176) and in each of the figure captions and have ensured that it is clear in the text to which emission scenario the result refer. We have performed additional simulations based on RCP4.5 emissions. The resulting figures can be found in the supplement of the manuscript. Fig. S6-S8 show the development of reference evapotranspiration, global radiation, and temperature for RCP4.5 and RCP8.5; Fig. S9-S15 are the corresponding figures for RCP4.5 from Fig. 6-12 of the manuscript. Table S4 corresponds to Table 3 of the manuscript. In the text of the manuscript, the results for RCP4.5 are described in lines 388-391 for annual precipitation and the climatic water balance, and for seasonal changes in lines 433-439. Regarding drought stress, the results for RCP4.5 are described in lines 502-508.

Concerning the mentioned study of Hübener et al. (2017) and the statement 'Noteworthy, the projected bandwidth for precipitation for the mitigation scenario RCP2.6 are less than half of those for RCP8.5 (Hübener et al., 2017).' the findings of the study are compared in lines 554-556.

The role of RCP8.5, as the most extreme emission scenario, is discussed in lines 578-585.

2.6 Statistical interpretation

There is no detailed interpretation of the p-value trends shown in Figures 6b and 7b, only the brief statements:

Lines 290-1: 'For seven simulations, the projected trends were significant after the year 2073 (Mann–Kendall trend test, $p < 0.05$, Fig. 6b).'

Lines 292-3: 'The statistical significance of the trends was comparable to the trends of precipitation (Fig. 7b).'

Presumably, the null hypothesis being tested is that predicted precipitation trends are no different from zero, but the trends in p-values for the 10 models for both annual precipitation and climate water balance are only very briefly discussed. It seems to me that until about 2030 most models show no trend in precipitation, while by about 2070 eight out of ten models appear to show a statistically significant trend (in a couple of cases a negative trend). A major shift seems to take place between about 2030 and 2050. In contrast, the results for the climate water balance shown in Figure 7 seem markedly different, with only two models showing a statistically significant trend by about 2030, and much less agreement between models as to future trends. It would be useful to have further discussion of likely mechanisms here (in Section 3.3.1).

**Response:** Thank you for pointing this out. The contrast between the precipitation trends (Fig. 6) and trends of the climate water balance (Fig. 7) is related to the increase in reference evapotranspiration (see Table 3, line 337). We agree to discuss this more detailed.

**Changes in the manuscript:** We have revised the text in section 3.3.1 and discussed the results with more detail (lines 365-391), especially with regard to changes in weather variables driving changes in reference evapotranspiration. Minor changes have been made to figures 6 and7, where baselines have been added.

Similarly, it would be useful to have more critical analysis of the results shown in Figure 10. The remarkable difference between the potential drought stress for the two periods (1989-2018 and 2041-2070) is not adequately explained. Presumably, the wide range of values shown for 2041-2070 for both regions could be explained by three poor-performing models, and if they were removed the differences between 1989-2018 and 2041-2070 may actually be minimal (but there is no such critical analysis here). There are some rather vague qualitative comparisons of 'bandwidth' in modelled precipitation mentioned in Section 4 (Discussion), and in relation to model evaluation, it is stated that 'This bandwidth could be reduced if the extreme models at the upper or lower edge would be excluded, but since no direct model flaws were detected, this would exclude possible future climate realisations.' (Line 411-12). However, based on the information provided in the paper there does not seem to have been any serious attempt to undertake model validation (and I am not sure what a 'direct model flaw' is). There therefore seems to have been a lack of detailed critical analysis of the rather dated climate models used in this study, as mentioned earlier, and this seems to be a major weakness of this work.

**Response:** We focused on describing the uncertainty of the ensemble in terms of possible future climate developments rather than discussing individual models of the ensemble. A serious validation of the climate models, which would maybe end up in an exclusion of a climate simulation, is critical and this could not be performed for the reasons of feasibility mentioned above. What we meant with "direct model flaw" could also refer to errata of climate data, usage restrictions or other reported issues. Those issues could lead to an exclusion or withdrawal of a model in a project, for instance as reported by the Reklies-De project (see point 4. of http://reklies.wdc-climate.de/ but only in German). We suggest

formulating "severe shortcomings" instead of "direct model flaw". To the knowledge of the authors such shortcomings regarding the used climate models were not reported elsewhere.

**Changes in the manuscript:** We changed the formulation in line 548-550.

2.7 Lack of future research directions

There is no clear statement in the discussion outlining where this research might lead and what topics would be worth following up.

**Response:** We will add more information concerning this point, for both, the viticultural as well as the climate modelling perspective.

**Changes in the manuscript:** Concerning possible future research topics, we made changes to section 4.4 and described viticultural research questions in the field of adaptation measures to climate change. At the end of section 4.4 also research topics concerning climate modelling at a local level are described (lines 676-683).

3. **Technical corrections, including typing errors and English expression**

There are a lot of problems with basic English expression which in some cases make the explanations confusing. Some suggested changes are indicated below:

**Response:** We appreciate all corrections and will incorporate them into the revised manuscript.

**Changes in the manuscript:** All corrections were incorporated (but not marked-up). 'Bandwidth' is mostly replaced with 'range' (suggested by referee#2) and 'variability'.

**Referee #2:**

Comments on " Downscaling of climate change scenarios for a high resolution, site-specific assessment of drought stress risk for two viticultural regions with heterogeneous landscapes" by Hofmann et al.

Overall, this paper represents an advancement in our knowledge of the impact of climate change on viticulture albeit in this case for two very specific regions in Germany. Specifically, the authors incorporate site specific soil information into a vineyard water balance model to assess drought stress against the backdrop of climate change. I recommend acceptance subject to the following revisions.

Major Concerns:

1. While the paper focuses on drought stress, I strongly urge the authors to consider heat stress as well.  They have all the information at hand, so it should be relatively straight forward to consider in tandem both drought and heat stress. Moreover, the authors should go into more detail as to what specifically is driving ET changes in the future. Right now the description is rather vague between temperature and solar radiation.

**Response:** Thank you for this suggestion. Heat stress is a very interesting topic, but we think that a different methodology would be needed to address this topic in its entirety. Estimating heat stress on an individual vineyard basis would require the coupling of drought stress, stomatal closure, and the resulting changes in canopy energy balance. We opted to add this component in the future. In a first step, we wanted to identify possible "drought hot spots" because of the large heterogeneity of the terrain. Additionally, a different downscaling or bias correction method of the climate simulations would probably be needed for this type of analysis. The weather generator we used is well suited for reproducing the statistical structure of observed long time (30 years) weather recordings but less suited for reproducing frequencies of extremes, like record-breaking temperature events. In addition, up to now, heat stress is not a common stress factor of viticulture in Germany, even though hot days (Tmax > 30 °C) have been observed to increase in summer. The highest temperature recorded in Geisenheim (Rheingau, since July 1884) was 39.4 °C (on July 25th 2019). As stated above, we would need a refined energy balance model which would also need to include the energy balance of the soil and realistically, this would require additional validation runs a.s.o and would exceed the scope of the paper.

Concerning the question what is specifically driving ET changes, we would add this to the revised paper.

**Changes in the manuscript:** For the reasons described in the comment above, heat stress was not further investigated in the revised paper. Concerning the question, what is driving the changes in reference evapotranspiration, we analysed observed weather data and the climate projections. The results are shown in the supplement in Table S2 (observed data) and Table S3 (climate projections). The results are mentioned at the relevant places in the manuscript, as described in more detail in the following comments.

2.  There is no consideration of the importance of rooting depth on the water balance calculation. I see this as a potential serious deficiency. Given the wide range of soil types and the lack of irrigation, what is the range of root depths across the region? How sensitive are the calculations to vine age/rooting depth? At a minimum a sensitivity analysis should be performed for a realistic range of root depths and not just some average value.

**Response:** The calculations were made on the assumption that grapevine roots have access to the full available water capacity (AWC). These data are available up to a depth of 2.0 m for the entire regions. If the rooting depth is limited because of shallow soils for example, this would largely be reflected in the AWC data. It is known that grapevine roots can reach deep (> 6 m) soil layers, but 80 % of the roots are usually found within the upper 1.0 m (Smart et al., 2006). Many studies have shown that the water status of established grapevines (older than approximately 5-6 years) can be described quite well based on the water balance of the upper 1.5-2.0 m. Among these was also a study on vastly different vineyard sites within the experimental region (Hofmann et al., 2014). From this, we concluded that the available data on AWC was a proper estimate for the total transpirable soil water grapevines can maximally extract from the soil and we would outline this more clearly in the manuscript.

Young grapevines, especially in the first three years, would need an additional investigation. Yet, since vineyard renewal is in the order of 30-45 years, the proportion of surface area falling in this category would be between 6-10 %. We propose to add more details concerning the

rooting depth in the revised manuscript. We could add a sensitivity analysis for cases of concern, i.e. individual vineyards with low available AWC and varying rooting depth, but it would be difficult to apply to the situation of the region.

Smart, D. R., Schwass, E., Lakso, A., and Morano, L.: Grapevine Rooting Patterns: A Comprehensive Analysis and a Review, Am J Enol Viticult, 57, 89-104, 2006.

**Changes in the manuscript:** We have added the assumptions made on the rooting depths in section 2.4, lines 235-240 and added a sensitivity analysis in the supplement, Fig. S4.

Specific comments.

1. Lines, 20, 79, 86, etc. Why were the Rheingau and Hessiche Bergstrasse chosen for this study? I can understand the Rheingau as one of the world's most renowned wine regions, but why Hessiche Bergstrasse versus say the Mosel, the Pfalz or the Nahe? The reasoning as stated is not very convincing.

**Response:** The study was funded by the state government (Hesse) to which both wine regions belong. Also, the density and depth of available soil data is unique in Germany. Incorporating other wine regions would be very interesting but would be a topic for the future.

**Changes in the manuscript:** We have not made changes to the manuscript on this question; due to the more technical background.

2. Line 50, canopy management is mentioned in passing here as a possible mitigation for climate change but never really followed up in the discussion at the end.

**Response:** We agree to take up this topic again in the discussion.

**Changes in the manuscript:** Canopy management is now discussed in section 4.4, lines 653-660).

3. Please elaborate in more detail on issues with respect to access to water (e.g. the steep slopes) and water restrictions (e.g., appellation constraints).

**Response:** We agree to this suggestion and we would add more details concerning the access to water.

**Changes in the manuscript:** We have added the information that irrigation is allowed in Germany since 2002 (so there are no appellation constraints) in line 76. Concerning the access to water, we discussed some points in section 4.4, lines 661-666.

4. Line 86. There should be a fuller discussion as to the limits and uncertainties of downscaling at this scale.

**Response:** We agree that at this scale, characteristics such as climatic differences within an area covered by gridded climate model data become apparent, which should be discussed with more detail.

**Changes in the manuscript:** We have added more information concerning the downscaling methodology to section 2.2.2 and section 2.4, in particular with the reference to the flow diagrams S1 and S2 in the supplement. We moved the information of the spatial resolution of the soil data from the discussion (line 535) to section 2.4 (line 235). The limits and uncertainties of the downscaling approach, are now more clearly discussed in the section about the validation of the weather generator (section 3.1 lines 259-294).

5. Line 107, what region?

**Response:** This refers to the Rheingau region, we would add this to the revised manuscript.

**Changes in the manuscript:** We have clarified the region information (lines 110-111).

6. Page 4, bottom. Of the 10 weather stations considered, how many unique 25km RCM grid boxes are used?

**Response:** We used four grid boxes for the Rheingau and one for the Hessische Bergstraße. We would add this information to a more detailed description of the downscaling methodology (section 2.2.2) also suggested by Reviewer#1.

**Changes in the manuscript:** We have to correct the response above. Four grid boxes covered all stations in the Rheingau and one grid box the stations in the Hessische Bergstraße. However, in detail, the four closest grids were used for each station and the RCM-based climate change scenarios were interpolated into station data based on inverse distance weighting. This is now mentioned in the flow diagram S1 in the supplement.

7. Line 160, it is not clear at all why the annual mean global temperature of MAGICC is used and not the annual mean global temperature of the GCM climate change projections?

**Response:** To use the mean global temperature of the GCM climate change projections, it would have been necessary to calculate this for each GCM (including calculating the mean of all grid boxes covering the earth of the GCM). The reduced complexity model of MAGICC greatly simplifies this approach.

**Changes in the manuscript:** For clarification, we have added a graph concerning the scaling factor *k* in the supplement (Fig. S2), which depends on development of the global mean temperature of MAGICC and referenced to Fig. S2 in line 175.

8. Section 2.4: Again what is the sensitivity of these results as a function of root depth resulting from vine age, soil type, and site-specific water availability? What is the range of root depths across the region?

**Response:** We would add the assumptions made on the rooting depths to section 2.4.

**Changes in the manuscript:** As mentioned above, we have added the assumptions made on the rooting depths in section 2.4, lines 235-240 and added a sensitivity analysis in the supplement, Fig. S4.

9. Section 2.5: There is a missed opportunity here to also consider heat stress sensitivity when you have all the data at hand to so.

**Response:** Thank you for this suggestion. Please refer to our comments at the beginning of this response under the section – major concerns.

10. Section 3. The paper would benefit from more discussion at the end as to non-stationarity.

**Response:** This is a valid point we would follow up with more detail in the discussion. One possible already observed non-stationarity effect is mentioned in line 501-503 ("Due to increased temperature combined with relatively unchanged but still highly variable precipitation patterns (Fig. 8c), the occurrence of warm and wet conditions during the ripening period (September, October) has increased the risk for rot (Schultz and Hofmann, 2015)."). Non-stationarity effects could also be a point for future research directions.

**Changes in the manuscript:** We have followed up this point now in the discussion in the lines 670-676.

11. Lines 232-33. Per 10 above, what does this sentence mean? Why are extreme events underestimated?

**Response:** The weather generator is capable to reproduce the statistical structure of long-time observational weather but underestimates frequencies of years with high solar radiation and high reference evapotranspiration. We will clarify the point in the revision.

**Changes in the manuscript:** We revised section 3.1 (see the next comment) and better explained, why extreme events were underestimated.

12. Figure 2. Please show a power spectrum of the 30-year results not just the seasonal cycle and whisker plots.

**Response:** We agree to this suggestion. Reviewer#1 also suggested improving the statistical analyses.

**Changes in the manuscript:** We have rewritten section 3.1 completely (lines 258-294). In addition to the comparisons of seasonal and annual distributions in Fig. 2, we have added distributions of daily data, power spectra and an indirect validation by comparing water balance calculations of observed and synthetic data of the weather generator for three different weather stations. We think, that we have now presented the strength and weaknesses of the approach in a more comprehensible way.

13. Line 287. Please replace this and all subsequent uses of the word bandwidth. The most common use of bandwidth refers to a frequency range, for example when filtering a time series. Spread or range are much better.

**Response:** Thank you for noting this. We will correct this in the revised manuscript.

**Changes in the manuscript:** We have mostly replaced 'bandwidth' with 'range' and 'variability' (suggested be referee#1).

14. Lines 320-325, please detail what is driving these ET changes.

**Response:** We agree to analyse this with more detail and will discuss the causes in the manuscript.

**Changes in the manuscript:** We have detailed the driving ET changes in Table S3 in the supplement and referred to this results in section 3.3.1, lines 384-387.

15. Much like 13, please describe in greater detail what is mean by "ensemble change"

**Response:** We mean the spread of the ensemble. This will be corrected in the revised manuscript.

**Changes in the manuscript:** We have replaced 'bandwidth' with 'range' in the caption of Table 3, which should explain, what 'ensemble change' mean.

16. Line 429, please tease out what is driving ET

**Response:** We will add the individual contributions of the weather variables to ET changes.

**Changes in the manuscript:** This point in the manuscript refers to observed data. We have added a table in the supplement, where the contributions of the weather variables to the increase in ET is assessed and referred to this table (Table S2) in line 569.

17. Line 435, all the more reason to also include heat stress in this study not just drought stress

**Response:** Thank you again for pointing this out.

18. Line 501, please also discuss the potential role of canopy management

**Response:** We agree that canopy management, as potential adaptation strategy, should be discussed.

**Changes in the manuscript:** The potential role of canopy management is now discussed in section 4.4, lines 654-659.

---

## Referee Report (RR1)

**Review: Downscaling of climate change scenarios for a high resolution, site–specific assessment of drought stress risk for two viticultural regions with heterogeneous landscapes**

This paper presents a summary of a study focused on the effects of climate change on two viticultural regions in west-central Germany. Methods used include downscaling of Coupled Model Intercomparison Project 5 (CMIP5) using a stochastic weather generator data, high resolution soil map data (soil type and water capacity), high resolution digital elevation model, scaling down to individual vineyards (accounting for slope steepness and aspect). For validation purposes, the study compared historical weather observations from nearby long-term weather stations with historical synthetic time series.

**General comments**: The manuscript needs some editing for grammar and usage (e.g., abstract 2nd and 3rd sentences are unclear and confusing). There is little discussion of relative humidity, a critical component of any evapotranspiration discussion (ET ~ f(T, RH, U)), other than reference to Supplement Table S3, which shows little change in future RH—that in itself is interesting, given the sensitive of RH to changes in temperature. (How future climate change trends maintain an equilibrium in RH is an interesting find in itself.) Table S3 is rather confusing—based upon the Table caption it appears these values represent the sensitivity of ET to changes in one variable while maintaining others at baseline values. It would provide more insight if the authors explored changes in conservative variables such as specific humidity (or vapor pressure and saturation vapor pressure) and examined projected changes in the vapor pressure deficit (not mentioned at all in the manuscript). Overall, the study presents interesting and valuable results regarding the potential effects of climate change on viticulture in Germany. The study's techniques are transferable to other viticultural (agricultural) regions providing similar data are available for input.

**Specific comments**:
**Page 1, line 27** (first line of Introduction): this sentence is confusing as the actual language in the WMO report cited here states that "[s]ince the 1980s, each successive decade has been warmer than any preceding one since 1850."
**Page 2, line 67**: Need to spell out ADVICLIM.
**Page 4, line 159**: Same for ENSEMBLES.
**Page 5:** Figure 1 would be helpful if you can show the wine growing regions as an inset to a larger map of Germany.
**Page 6, lines 206 - 208:** "The impact of degree of slope on runoff was neglected, because several authors reported no clear findings…." Would this be the case during periods of drought (harder soil surface)?
**Page 13, lines 387 - 388**: "In general, this indicates an increase of precipitation in winter *possibly connected* with a decrease of precipitation in a future summer." (Italics added.) Why would an increase in winter precipitation be connected with a decrease in summer precipitation?
**Page 13, general comment**: What about snowfall and runoff from snow melt? Is this an issue in this region?

**Page 17 and 18, Figures 11 and 12**: difficult to see the changes for the ensemble median (b) and decrease (c) as changes are generally modest compared with the scale (a function of the larger changes exhibited in (a).

**Page 19, lines 498-499**: "… but in general, the soil maps are still describing the current situation quite well as demonstrated in a follow–up study (Zimmer, 1999)." This study is more than 20 years old.

**Page 19, lines 520 et seq**.: under the more extreme scenario, is the increase in the number of predicted drought days exceed any year in the past? This would also be a good place to look at changes in vapor pressure deficit, a key control on evapotranspiration (e.g., Penman-Monteith— see Monteith and Unsworth 1990 2nd Ed.)

---

## Author Response (AR2)

**Review: Downscaling of climate change scenarios for a high resolution, site–
specific assessment of drought stress risk for two viticultural regions
with heterogeneous landscapes**

This paper presents a summary of a study focused on the effects of climate change on two viticultural regions in west-central Germany. Methods used include downscaling of Coupled Model Intercomparison Project 5 (CMIP5) using a stochastic weather generator data, high resolution soil map data (soil type and water capacity), high resolution digital elevation model, scaling down to individual vineyards (accounting for slope steepness and aspect). For validation purposes, the study compared historical weather observations from nearby long-term weather stations with historical synthetic time series.

The authors thank the referee for the time and attention spent on the review and the helpful comments. Our answers to the comments are given in blue text colour.

**General comments:** The manuscript needs some editing for grammar and usage (e.g., abstract 2nd and 3rd sentences are unclear and confusing). There is little discussion of relative humidity, a critical component of any evapotranspiration discussion (ET ~ f(T, RH, U)), other than reference to Supplement Table S3, which shows little change in future RH—that in itself is interesting, given the sensitive of RH to changes in temperature. (How future climate change trends maintain an equilibrium in RH is an interesting find in itself.) Table S3 is rather confusing—based upon the Table caption it appears these values represent the sensitivity of ET to changes in one variable while maintaining others at baseline values. It would provide more insight if the authors explored changes in conservative variables such as specific humidity (or vapor pressure and saturation vapor pressure) and examined projected changes in the vapor pressure deficit (not mentioned at all in the manuscript). Overall, the study presents interesting and valuable results regarding the potential effects of climate change on viticulture in Germany. The study's techniques are transferable to other viticultural (agricultural) regions providing similar data are available for input.

**Response**: We agree that relative humidity need to be more discussed in the context of evapotranspiration and have added more details about the changes of observed and projected relative humidity in the discussion part. We added results of the vapour pressure deficit to Tables S1-S3 and a note to Table S3 to improve clarity.

**Specific comments:**

**Page 1, line 27** (first line of Introduction): this sentence is confusing as the actual language in the WMO report cited here states that "[s]ince the 1980s, each successive decade has been warmer than any preceding one since 1850."

**Response**: Thank you for this advice, we have adapted the sentence to fit the statement of the WMO report.

**Page 2, line 67**: Need to spell out ADVICLIM.

**Response**: The meaning of the acronym ADVICLIM is described in the reference (Quénol et al., 2014), so for the reason of readability we propose to use only the name of the project here.

**Page 4, line 159**: Same for ENSEMBLES.

**Response**: To our knowledge, ENSEMBLES is the name of the project and cannot be spelled out in more detail. As we mentioned the reference of the project (van der Linden and Mitchell, 2009), we suggest not to change the sentence.

**Page 5**: Figure 1 would be helpful if you can show the wine growing regions as an inset to a larger map of Germany.

**Response**: We added a map of Germany with the region of Figure 1 as an inset to Figure 1.

**Page 6, lines 206 - 208**: "The impact of degree of slope on runoff was neglected, because several authors reported no clear findings…." Would this be the case during periods of drought (harder soil surface)?

**Response**: The curve number method (surface runoff model) accounts for the "antecedent moisture condition" of a soil before a precipitation event. Since the degree of slope does not necessarily reduce the infiltration capacity, we assume neglecting the impact of slope on runoff is also valid for periods of drought.

**Page 13, lines 387 - 388**: "In general, this indicates an increase of precipitation in winter *possibly connected* with a decrease of precipitation in a future summer." (Italics added.) Why would an increase in winter precipitation be connected with a decrease in summer precipitation?

**Response**: Indeed the formulation is misleading and we have revised the sentence and removed the word "connected".

**Page 13**, general comment: What about snowfall and runoff from snow melt? Is this an issue in this region?

**Response**: Snow melt is not really an issue and with global warming snowfall has decreased. In the region snowfall appears only on some days during winter and in general snow melts quickly. Snow covers with more than 10 cm are rare and do not occur every winter.

**Page 17 and 18, Figures 11 and 12**: difficult to see the changes for the ensemble median (b) and decrease (c) as changes are generally modest compared with the scale (a function of the larger changes exhibited in (a).

**Response:** Yes, the colour scale is a compromise between showing the modest changes without overemphasizing them in comparison with the results of warm simulation in (a).

**Page 19, lines 498-499**: "… but in general, the soil maps are still describing the current situation quite well as demonstrated in a follow–up study (Zimmer, 1999)." This study is more than 20 years old.

**Response:** The soil characteristics have not changed in general in the regions since the study of Zimmer from 1999, beside of some locally small-scale interventions. So we think the statement is still valid.

**Page 19, lines 520 et seq.**: under the more extreme scenario, is the increase in the number of predicted drought days exceed any year in the past? This would also be a good place to look at changes in vapor pressure deficit, a key control on evapotranspiration (e.g., Penman-Monteith— see Monteith and Unsworth 1990 2nd Ed.)

**Response:** Simulated drought stress days with observed weather data for the dry year 2018 show up to 90 days of drought stress, with many vineyards in the range of 50-90 days (see Fig. 5 on page 11). As the projections show the difference of mean annual drought stress days between 2041-2070 and 1989-2018 with many vineyards showing an increase of drought stress days in the range 33-69 days, this would mean that many years in future are in the range of the dry year 2018 and some years will likely exceed the dry year 2018.
We added the discussion about relative humidity and vapour pressure deficit at this point in the discussion part as suggested by the referee.